# Variational Phylogenetic Inference with Products over Bipartitions

Evan Sidrow [1]   Alexandre Bouchard-Côté [2]   Lloyd T. Elliott [1]

## Abstract

Bayesian phylogenetics is vital for understanding evolutionary dynamics, and requires accurate and efficient approximation of posterior distributions over trees. In this work, we develop a variational Bayesian approach for ultrametric phylogenetic trees. We present a novel variational family based on coalescent times of a single-linkage clustering and derive a closed-form density for the resulting distribution over trees. Unlike existing methods for ultrametric trees, our method performs inference over all of tree space, it does not require any Markov chain Monte Carlo subroutines, and our variational family is differentiable. Through experiments on benchmark genomic datasets and an application to the viral RNA of SARS-CoV-2, we demonstrate that our method achieves competitive accuracy while requiring significantly fewer gradient evaluations than existing state-of-the-art techniques.

## 1. Introduction

The goal of Bayesian phylogenetics is to infer the genealogy of a collection of taxa given a genetic model and aligned sequence data. Phylogenetics is used in fields such as epidemiology (Li et al., 2020), linguistics (List et al., 2014), and ecology (Godoy et al., 2018). Bayesian phylogenetic inference quantifies uncertainty and integrates over phylogenetic tree structures within a phylogenetic model (Zhang and Matsen IV, 2019). Most Bayesian phylogenetic inference is performed using Markov chain Monte Carlo (MCMC) methods with candidate trees iteratively proposed and either accepted or rejected based on their consistency with the observed data. However, MCMC methods can struggle because the number of possible trees grows super-exponentially in the number of taxa and posteriors on trees are highly multi-modal.

One alternative to MCMC is variational inference (VI), in which the posterior distribution over phylogenetic tree structures is approximated using a variational distribution that minimizes some distance metric to the true posterior distribution. While VI over combinatorial spaces is also known to be difficult due to the complexity of the constraints on the support (Bouchard-Côté and Jordan, 2010; Linderman et al., 2018), there have been several recent advances in VI over phylogenetic trees that are tractable. Zhang and Matsen IV (2018) represented phylogenetic trees as Bayesian networks using *subsplit Bayesian networks* (SBNs), and later used SBNs to perform variational Bayesian phylogenetic inference on unrooted trees (Zhang and Matsen IV, 2019). This approach has spawned many methodological advancements. To improve the distribution for branch lengths, Zhang (2020) used normalizing flows, Molén et al. (2024) used mixtures, and Xie et al. (2024) used semi-implicit branch length distributions. To improve the variational family over tree topologies, Zhang (2023) used graph neural networks to learn topological features.

The number of parameters within an SBN grows exponentially with the number of taxa, so Zhang and Matsen IV (2019) used MCMC to find sets of most likely tree structures, and they use these sets to restrict the SBNs. Other recent VI approaches sample tree topologies without the use of SBNs. For example, ViaPhy (Koptagel et al., 2022) uses a gradient-free variational inference approach and directly sample from the Jukes and Cantor (1969) model, GeoPhy (Mimori and Hamada, 2023) uses a distance-based metric in hyperbolic space to construct unrooted phylogenetic trees, and ARTree (Xie and Zhang, 2023) uses graph neural networks to construct a deep autogressive model for VI over phylogenetic tree structures. Zhou et al. (2024) also introduce PhyloGFN, a phylogenetic VI technique based on reinforcement learning and generative flow networks (Bengio et al., 2023).

Phylogenetic inference can also be performed using non-Bayesian methods, including RAxML (Stamatakis, 2014), neighbour-joining (Saitou and Nei, 1987), and more recently Phyloformer (Nesterenko et al., 2025). Phyloformer uses deep learning to construct pairwise representations of evolutionary distances between taxa. Phyloformer then uses pairwise distances to construct a tree using a neighbour-joining algorithm similar to the single-linkage clustering

---

[1]Department of Statistics and Actuarial Science, Simon Fraser University, Burnaby, Canada [2]Department of Statistics, University of British Columbia, Vancouver, Canada. Correspondence to: Evan Sidrow <esidrow@sfu.ca>.

*Proceedings of the 42$^{nd}$ International Conference on Machine Learning*, Vancouver, Canada. PMLR 267, 2025. Copyright 2025 by the author(s).

algorithm described here. However, non-Bayesian methods do not provide estimates of marginal likelihood, which are useful for model selection.

In this work we focus on rooted ultrametric phylogenetic trees, for which branch lengths correspond to the amount of time between evolutionary branching events. This formulation is useful when time is important, for example in applications involving rapidly evolving pathogens (Sagulenko et al., 2018). None of the aforementioned approaches incorporate time constraints into the branch lengths of the phylogenetic trees. To this end, Zhang and Matsen IV (2024) generalized their SBN-based approach to ultrametric trees, but they still rely on MCMC to restrict to a subset of tree space upon which to perform inference over. Alternatively, Bouckaert (2024) provides cubeVB, a method related to the one described in our manuscript. However, because the matrix representation of tree space in Bouckaert (2024) is not dense, they can only express a limited number of trees. As such, their variational family is not supported on many tree topologies. Further, they do not perform optimization on the tree structure, and instead rely on empirical values derived from MCMC.

We introduce **V**ariational phylogenetic **I**nference with **PR**oducts over bipartitions (VIPR), a new variational family for ultrametric trees based on coalescent theory and single-linkage clustering (Kingman, 1982). VIPR naturally performs variational inference on ultrametric trees and thus directly incorporates time into phylogenetic inference. VIPR also performs inference over the entirety of tree space and does not rely on MCMC subroutines. In particular, we parameterize a variational distribution over a distance matrix and use it to derive a differentiable variational density over trees that result from single-linkage clustering. Through a set of experiments on standard datasets and an application to COVID-19, we show that our simple variational formulation achieves comparable results to existing methods for ultrametric trees in fewer gradient evaluations.

## 2. Background

Consider a set of $N$ taxa $\mathcal{X} = \{x_1, x_2, \ldots, x_N\}$. A nonempty subset $X$ of $\mathcal{X}$ is referred to as a *clade* of $\mathcal{X}$. A clade represents a collection of taxa which share a common ancestor at a particular time in the past. Further, we represent an evolutionary branching events using a bipartition $\{W, Z\}$ of the clade $X$ ($X = W \cup Z$, $W \cap Z = \emptyset$).

We focus on *ultrametric trees*, in which the leaves of the trees are all equidistant from the root. We denote an ultrametric tree with a rooted, binary tree topology $\tau$ and a set of coalescent times $\mathbf{t} = \{t_n\}_{n=1}^{N-1}$, where there is one $t_n$ for each internal node in $\tau$. For ultrametric trees in particular, the branch length between a child and its parent is equal to

the difference in coalescent times between the parent and the child nodes. We measure $\mathbf{t}$ in backwards time, so each $t_n$ is positive and represents a time before the present. The leaves of $\tau$ correspond to the genomes of each measured taxon $x \in \mathcal{X}$. Additionally, an internal node $u$ of $\tau$ represents the (unobserved) genome of the most recent common ancestor of all taxa that have $u$ as a parent node. As a binary tree, $\tau$ contains a total of $N - 1$ internal nodes (including the root node). We can thus represent the tree with a collection of bipartitions: $\tau = \{\{W_n, Z_n\}\}_{n=1}^{N-1}$. In this representation, an internal node $u_n$ is the most recent common ancestor for all taxa $x \in W_n \cup Z_n$, and the $n$-th coalescent event is represented by the bipartition $\{W_n, Z_n\}$.

Denote the set of possible characters within a set of aligned genetic sequences by $\Omega$ (*e.g.*, a DNA sequence may correspond to $\Omega = \{A, T, G, C\}$ and an RNA sequence to $\Omega = \{A, U, G, C\}$). Further, denote the set of observed genomes by $\mathbf{Y}^{(\text{ob})} = \{Y_1^{(\text{ob})}, \ldots, Y_M^{(\text{ob})}\}$, where $Y_m^{(\text{ob})} = (Y_{m,x_1}, \ldots, Y_{m,x_N})$ corresponds to the base pairs at site $m$ for all observed taxa $x \in \mathcal{X}$. In addition to the observed genomes $\mathbf{Y}^{(\text{ob})}$, denote the unobserved genomes of all internal nodes by $\mathbf{Y}^{(\text{un})} = \{Y_1^{(\text{un})}, \ldots, Y_M^{(\text{un})}\}$, where $Y_m^{(\text{un})} = (Y_{m,u_1}, \ldots, Y_{m,u_{N-1}})$ corresponds the base at site $m$ for all *unobserved* internal nodes $u_1, \ldots, u_{N-1}$. Let the index of the root node be $N - 1$ (so, $u_{N-1}$ is the root node). We denote the combined observed and unobserved genomes by $\mathbf{Y} = \{\mathbf{Y}^{(\text{ob})}, \mathbf{Y}^{(\text{un})}\}$. For further background on phylogenetics, we refer to Hein et al. (2004).

### 2.1. Phylogenetic Likelihood

For simplicity, we focus on the Jukes and Cantor (1969) model of evolution. We denote the stationary distribution by $\boldsymbol{\pi}$, and the transition matrix by $P(b)$ (a $4 \times 4$ matrix such that the $i, j$-th entry is the probability of transitioning from base $i$ to base $j$ given branch length $b$ under Jukes and Cantor 1969). With this notation, the likelihood of an observed set of genetic sequences $Y^{(\text{ob})}$ at site $m$ is as follows:

$$p(Y_m^{(\text{ob})} | \tau, \mathbf{t}) = \sum_{Y_m^{(\text{un})}} \boldsymbol{\pi}(Y_{m,r}) \prod_{(u,v)} (P(b_{u,v}(\tau, \mathbf{t})))_{Y_{m,u}, Y_{m,v}}.$$

Here the product is over all edges $(u, v)$ in $\tau$. We assume independence between sites, and so the likelihood of the observed genomes is:

$$p(\mathbf{Y}^{(\text{ob})} | \tau, \mathbf{t}) = \prod_{m=1}^{M} p(Y_m^{(\text{ob})} | \tau, \mathbf{t}). \qquad (1)$$

Equation (1) can be evaluated in $\mathcal{O}(NM)$ time using the pruning algorithm (Felsenstein 1981, also known as the sum-product algorithm; Koller and Friedman 2009).

## 2.2. Prior Distribution over Trees

We use the Kingman coalescent (Kingman, 1982) as the prior distribution on trees. This coalescent process proceeds backward in time with independent and exponentially distributed inter-event intervals. Events occur at rate $\lambda_k = \binom{k}{2}/N_e$. Here $k$ is the number of extant taxa and $N_e$ is the effective population size, a parameter which governs the rate at which taxa coalesce. We fix $N_e = 5$ in our experiments. At each event, a pair of extant taxa are chosen to coalesce into a single taxon uniformly at random over all pairs of extant taxa, yielding the prior:

$$p(\tau, \mathbf{t}) = \frac{2^{N-1}}{N!(N-1)!} \prod_{k=2}^{N} \lambda_k \exp\left(-\lambda_k(t_k - t_{k-1})\right).$$

## 2.3. Variational Inference for Phylogenetic Trees

Our goal is to infer a distribution over tree structures and coalescent times given observed genetic sequences:

$$p(\tau, \mathbf{t} \mid \mathbf{Y}^{(\mathrm{ob})}) = p(\mathbf{Y}^{(\mathrm{ob})} \mid \tau, \mathbf{t})\, p(\tau, \mathbf{t})/p(\mathbf{Y}^{(\mathrm{ob})}). \quad (2)$$

Here $p(\mathbf{Y}^{(\mathrm{ob})})$ is an intractable normalizing constant. Variational inference involves defining a tractable family of probability densities parameterized by some variational parameters $\phi$. Then, the posterior density is approximated by a variational density $q_\phi(\tau, \mathbf{t})$ whose parameters $\phi$ should minimize a divergence measure $D$ between the posterior $p(\cdot \mid \mathbf{Y}^{(\mathrm{ob})})$ and $q_\phi$. Here we use the reverse KL divergence:

$$D_{KL}(q_\phi \,\|\, p) = \mathbb{E}_{(\tau, \mathbf{t}) \sim q_\phi}\left[\log\left(\frac{q_\phi(\tau, \mathbf{t})}{p(\tau, \mathbf{t} \mid \mathbf{Y}^{(\mathrm{ob})})}\right)\right]. \quad (3)$$

Evaluating the exact posterior $p(\tau, \mathbf{t} \mid \mathbf{Y}^{(\mathrm{ob})})$ is difficult. Instead, we equivalently (up to a normalizing constant) maximize the evidence lower bound (ELBO):

$$\phi^* = \arg\max_{\phi} L(\phi). \quad (4)$$

$$L(\phi) = \mathbb{E}_{q_\phi}\left[\log\left(\frac{p(\tau, \mathbf{t}, \mathbf{Y}^{(\mathrm{ob})})}{q_\phi(\tau, \mathbf{t})}\right)\right]. \quad (5)$$

ELBO is also known as the negative variational free energy in statistical physics and some areas of machine learning. The expectation over $q_\phi$ consists of a sum over tree structures $\tau$ and an integral over coalescent times $\mathbf{t}$, forming the following objective function:

$$L(\phi) = \sum_{\tau} \int_{\mathbf{t}} q_\phi(\tau, \mathbf{t}) \log\left(\frac{p(\tau, \mathbf{t}, \mathbf{Y}^{(\mathrm{ob})})}{q_\phi(\tau, \mathbf{t})}\right) d\mathbf{t}. \quad (6)$$

---

**Algorithm 1** Single-Linkage Clustering$(\mathbf{T}, \mathcal{X}_0)$

1: **Input:** Distances $\mathbf{T} \in \mathbb{R}_{>0}^{\binom{N}{2}}$ and taxa set $\mathcal{X}_0 = \{\{x_1\}, \{x_2\}, \dots, \{x_N\}\}$.

2: **for** $n = 1, \dots, N-1$ **do**

3: $\quad w^*, z^* \leftarrow \arg\min_{w,z}\{t^{\{w,z\}} : w, z \text{ not coalesced}\}$

4: $\quad$ Set $W_n \in \mathcal{X}_0$ to be the set containing $w^*$

5: $\quad$ Set $Z_n \in \mathcal{X}_0$ to be the set containing $z^*$

6: $\quad t_n \leftarrow t^{\{w^*, z^*\}}$

7: $\quad$ Remove $W_n, Z_n$ from $\mathcal{X}_0$ and add $W_n \cup Z_n$ to $\mathcal{X}_0$

8: **end for**

9: $\tau \leftarrow \{\{W_n, Z_n\}\}_{n=1}^{N-1}$

10: $\mathbf{t} \leftarrow \{t_n\}_{n=1}^{N-1}$

11: **Return** $(\tau, \mathbf{t})$

---

## 2.4. Matrix Representation of Tree Space

One way to construct a phylogenetic tree is to use a distance matrix $\mathbf{T}$ (a symmetric $N \times N$ matrix with positive and finite off-diagonal entries) and the *single-linkage clustering* algorithm, as described in Algorithm 1. We denote the distance between taxa $u$ and $v$ by $t^{\{u,v\}}$ and formulate the algorithm to return a representation of a phylogenetic tree using bipartitions and coalescent times that is consistent with our notation. We consider the distance matrix as an element of $\mathbb{R}_{>0}^{\binom{N}{2}}$ by identifying the off-diagonal elements.

Algorithm 1 is a naïve implementation of single-linkage clustering with time complexity $\mathcal{O}(N^3)$ and space complexity $\mathcal{O}(N^2)$. Sibson (1973) introduce SLINK, an implementation of Algorithm 1 with time complexity $\mathcal{O}(N^2)$ and space complexity $\mathcal{O}(N)$, and prove that both are optimal.

Bouckaert (2024) introduce cubeVB, a method that uses single-linkage clustering to perform variational inference over ultrametric trees. First, they note that if exactly $N-1$ entries of $\mathbf{T}$ are finite, then single-linkage clustering implies a bijection between $\mathbf{T}$ and $(\tau, \mathbf{t})$. Then, they specify exactly $N-1$ entries of $\mathbf{T}$ to be random and finite, setting all other entries to infinity. Next, they run an MCMC algorithm over trees to obtain an empirical distribution over coalescent times. Finally, they estimate the parameters associated with the $N-1$ finite entries of $\mathbf{T}$ using the MCMC-generated empirical distribution. This method has two main drawbacks—all entries of $\mathbf{T}$ must be specified to form a distribution supported on the entire tree space, and it is unclear how to select the $N-1$ best entries of $\mathbf{T}$ to cover the most posterior probability. To address these drawbacks, we present a gradient-based variational inference method for ultrametric trees based on single-linkage clustering that specifies a variational distribution over the entire tree space.

$$\mathbf{T} = \begin{array}{c} \\ A \\ B \\ C \\ D \end{array} \begin{array}{cccc} A & B & C & D \\ \end{array} \left( \begin{array}{cccc} * & \mathbf{2} & 8 & \mathbf{4} \\ * & * & 4.5 & 7 \\ * & * & * & \mathbf{3} \\ * & * & * & * \end{array} \right), \quad \mathbf{T} = \begin{array}{cccc} A & B & C & D \\ \end{array} \left( \begin{array}{cccc} * & \mathbf{2} & 5 & 6 \\ * & * & \mathbf{4} & 7 \\ * & * & * & \mathbf{3} \\ * & * & * & * \end{array} \right),$$

Figure 1. ***Schematic showing the sampling process for VIPR***. *This diagram shows two possible example matrices* $\mathbf{T}$ *(on the left) that could be drawn using* $t^{\{u,v\}} \sim q_\phi^{\{u,v\}}$ *and result in the same phylogenetic tree* $(\tau, \mathbf{t}) \sim q_\phi$ *(on the right) after running single-linkage clustering. Entries of* $\mathbf{T}$ *that trigger a coalescence event are shown in bold. The form of* $q_\phi^{\{u,v\}}$ *is quite general and can be provided by the practitioner, while the expression for* $q_\phi$ *depends upon* $q_\phi^{\{u,v\}}$.

## 3. Methods

We now present our method **V**ariational phylogenetic **I**nference with **PR**oducts over bipartitions (VIPR). We begin by outlining a generative process for sampling from our variational distribution $q_\phi$. We then describe how to evaluate the density of our variational distribution. Finally, we describe the optimization procedure used to maximize our variational objective function.

### 3.1. Generative Process for Phylogenies

We begin by describing a generative process for sampling from $q_\phi$, as our variational distribution is best understood through the algorithm for sampling from it. Our algorithm to sample an ultrametric tree with leaf nodes $\mathcal{X}$ proceeds similarly to Algorithm 1 of Bouckaert (2024). Namely, we randomly draw each element of the distance matrix $\mathbf{T}$ ($t^{\{u,v\}}$ for all $\{u,v\}$ with $u, v \in \mathcal{X}$) using a set of independent variational distributions with densities $q_\phi^{\{u,v\}}$. Then, we run single linkage clustering on $\mathbf{T}$ to form $(\tau, \mathbf{t})$.

---

**Algorithm 2** Sample-q($\boldsymbol{\mu}, \boldsymbol{\sigma}, \mathcal{X}$)

1: **Input:** Parameters $\boldsymbol{\mu} \in \mathbb{R}^{\binom{N}{2}}$ and $\boldsymbol{\sigma} \in \mathbb{R}^{\binom{N}{2}}_{>0}$ and taxa set $\mathcal{X} = \{\{x_1\}, \{x_2\}, \ldots, \{x_N\}\}$.
2: Draw $z^{\{u,v\}} \sim \mathcal{N}(0, 1)$ for all $\{u, v\} \subset \mathcal{X}$
3: Define matrix $\mathbf{T} \in \mathbb{R}^{\binom{N}{2}}_{>0}$ such that:
$$\log\left(t^{\{u,v\}}\right) = \mu^{\{u,v\}} + z^{\{u,v\}}\sigma^{\{u,v\}}$$
4: **Return** Single-Linkage Clustering($\mathbf{T}, \mathcal{X}$)

---

Note that $q_\phi^{\{u,v\}}(t^{\{u,v\}})$ and $q_\phi(\tau, \mathbf{t})$ are closely related: $q_\phi^{\{u,v\}}$ describes the distribution over entry $t^{\{u,v\}}$ of $\mathbf{T}$, while $q_\phi$ describes the distribution over phylogenetic trees

$(\tau, \mathbf{t})$ formed by running single-linkage clustering on $\mathbf{T}$. Algorithm 2 presents pseudocode to sample from $q_\phi$ if $q_\phi^{\{u,v\}}$ is a log-normal distribution, while Figure 1 visualizes the process of drawing $\mathbf{T}$ using $t^{\{u,v\}} \sim q_\phi^{\{u,v\}}$ and then using single-linkage clustering to map $\mathbf{T}$ to $(\tau, \mathbf{t})$.

### 3.2. Density Evaluation

In this section we describe how to evaluate the density of trees generated from Algorithm 2. The primary challenge is that a given tree $(\tau, \mathbf{t})$ may have been generated from multiple distance matrices $\mathbf{T}$ (see Figure 1). Luckily, this sampling procedure still yields a density with a closed-form solution, as shown in Proposition 1 below.

**Proposition 1.** *If the random variables* $t^{\{u,v\}}$ *are mutually independent, and all* $q_\phi^{\{u,v\}}$ *are continuous in* $\phi$ *and* $t$ *for all* $\{u, v\}$ *with* $u, v \in \mathcal{X}$, *and* $Q_\phi^{\{u,v\}}$ *is the survival function of* $t^{\{u,v\}}$, *then* $q_\phi(\tau, \mathbf{t})$ *has the following form:*

$$q_\phi(\tau, \mathbf{t}) = \prod_{n=1}^{N-1} \left( \left( \sum_{\substack{w \in W_n \\ z \in Z_n}} \frac{q_\phi^{\{w,z\}}(t_n)}{Q_\phi^{\{w,z\}}(t_n)} \right) \prod_{\substack{w \in W_n \\ z \in Z_n}} Q_\phi^{\{w,z\}}(t_n) \right). \tag{7}$$

A derivation of Proposition 1 using induction is provided in Appendix A. Every taxa pair $\{u, v\}$ appears in the sum and product terms of Equation (7) exactly once, as each taxa pair coalesces exactly once within a rooted phylogenetic tree. Thus, evaluating both $q_\phi(\tau, \mathbf{t})$ and $\nabla_\phi \log q_\phi(\tau, \mathbf{t})$ takes $\mathcal{O}(N^2)$ time.

If $q_\phi^{\{u,v\}}$ is continuously differentiable, then $q_\phi$ is also continuously differentiable. In our VIPR implementation,

$q_\phi^{\{u,v\}}$ is log-normal, so we can compute gradients with respect to $\phi$. Note however that Proposition 1 holds for any continuous mutually independent $q_\phi^{\{u,v\}}$.

---

**Algorithm 3** $\mathrm{VIPR}(\mathcal{X}, K)$

---

1: **Input:** Integer $K$ indicating number of samples to use in gradient approximation and taxa set $\mathcal{X} = \{\{x_1\}, \{x_2\}, \ldots, \{x_N\}\}$.

2: Initialize variational parameters $\phi$

3: **while** not converged **do**

4:     **for** $k = 1, \ldots, K$ **do**

5:         Draw $\mathbf{T}^{(k)} \in \mathbb{R}_{>0}^{\binom{N}{2}}$ with $t^{\{u,v\}} \sim q_\phi^{\{u,v\}}$

6:         $\left(\tau^{(k)}, \mathbf{t}^{(k)}\right) \leftarrow$
            $\text{Single-Linkage Clustering}(\mathbf{T}^{(k)}, \mathcal{X})$

7:     **end for**

8:     Estimate gradient $\nabla_\phi L(\phi)$ using $\left(\tau^{(k)}, \mathbf{t}^{(k)}\right)$
        for $k = 1, \ldots, K$.

9:     Update $\phi$ using gradient estimates and a stochastic
        optimization algorithm (Adam, SGD, *etc.*)

10: **end while**

11: **Return** $\phi$

---

### 3.3. Gradient Estimators for $q_\phi$

We now have almost everything we need to perform phylogenetic variational inference: an (unnormalized) phylogenetic posterior density $p(\tau, \mathbf{t}, \mathbf{Y}^{(\text{ob})})$, a variational family with density $q_\phi(\tau, \mathbf{t})$, and an objective function $L(\phi)$ to maximize in order to find a variational posterior distribution. We collect these steps together in Algorithm 3. Note that in this algorithm, optimizing $L(\phi)$ with stochastic gradient methods such as Adam (Robbins and Monro, 1951; Kingma and Ba, 2014) requires random estimates of the gradient $\nabla_\phi L(\phi)$. Thus, we consider three methods for gradient estimation and use them in our experiments: leave-one-out REINFORCE (Mnih and Gregor, 2014; Shi et al., 2022), the reparameterization trick (Rubinstein, 1992; Kingma and Welling, 2014), and VIMCO (Mnih and Rezende, 2016). An overview of these methods are given in the remainder of this subsection, with details in Appendix C. Our code implementing VIPR is available at https://github.com/EvanSidrow/VIPR.

#### 3.3.1. THE REINFORCE ESTIMATOR

Define $f_\phi(\tau, \mathbf{t}) \equiv \log(p(\tau, \mathbf{t}, \mathbf{Y}^{(\text{ob})})) - \log(q_\phi(\tau, \mathbf{t}))$, so that $L(\phi) = \mathbb{E}_{q_\phi}[f_\phi(\tau, \mathbf{t})]$. We can interchange the gradient and the finite sum over $\tau$ in Equation (6), and we assume that we can interchange the gradient and integral (see L'Ecuyer 1995 for technical conditions). After performing some algebra (see Appendix C.1), we obtain the *leave-one-out RE-*

*INFORCE* (LOOR) estimator (Mnih and Gregor, 2014; Shi et al., 2022). The gradient $\nabla_\phi \log q_\phi(\tau^{(k)}, \mathbf{t}^{(k)})$ can be calculated using automatic differentiation software such as Autograd (Maclaurin et al., 2015) or PyTorch (Paszke et al., 2019).

#### 3.3.2. THE REPARAMETERIZATION TRICK

The push out estimator (Rubinstein, 1992) is popular in machine learning literature under the name of the *reparameterization trick* (Kingma and Welling, 2014). Recall that in our experiments $q_\phi^{\{u,v\}}$ is a log-normal distribution for all $u, v \in \mathcal{X}$. In Algorithm (2), the candidate coalescent times $t^{\{u,v\}} \sim \text{log-normal}(\mu^{\{u,v\}}, \sigma^{\{u,v\}}) \iff t^{\{u,v\}} = \exp(\mu^{\{u,v\}} + \sigma^{\{u,v\}} z^{\{u,v\}})$ with $z^{\{u,v\}} \sim \mathcal{N}(0,1)$. Denoting the set $\{z^{\{u,v\}}\}_{u,v \in \mathcal{X}}$ by $\mathbf{Z}$, we reparameterize the expectation in Equation (5) as follows:

$$L(\phi) = \mathbb{E}_{\mathbf{Z}} \left[ \log \left( \frac{p(\mathbf{Y}, g_\phi(\mathbf{Z}))}{q_\phi(g_\phi(\mathbf{Z}))} \right) \right]. \qquad (8)$$

Here $g_\phi(\mathbf{Z}) = \text{Single-Linkage Clustering}(\exp(\boldsymbol{\mu} + \boldsymbol{\sigma} \odot \mathbf{Z}), \mathcal{X})$. Denoting the density of a $\binom{N}{2}$-dimensional standard normal distribution as $\mathcal{N}(\cdot; \mathbf{0}, \mathbf{I})$, we have:

$$L(\phi) = \int_{\mathbf{Z}} \mathcal{N}(\mathbf{Z}; \mathbf{0}, \mathbf{I}) \log \left( \frac{p(\mathbf{Y}, g_\phi(\mathbf{Z}))}{q_\phi(g_\phi(\mathbf{Z}))} \right) d\mathbf{Z}. \quad (9)$$

Using this formulation and some additional algebra, in Appendix C.2 we derive an estimator for the full gradient $\nabla_\phi L(\phi)$ using random samples of $\mathbf{Z}$. Unfortunately, this estimator is biased because the gradient and integral cannot be interchanged without introducing some error (see Appendix C.2). Therefore, this optimization procedure is not guaranteed to converge to a local optimum of the objective function. Nonetheless, these gradient estimates tend to perform at least comparably to the LOOR estimator.

#### 3.3.3. THE VIMCO ESTIMATOR

One drawback of the single-sample ELBO in Equation (5) is that variational distributions that target the ELBO tend to be mode-seeking (*i.e.*, they can underestimate variance of the true posterior). As an alternative, Mnih and Rezende (2016) suggest a $K$-sample ELBO (*VIMCO*: variational inference for Monte Carlo objectives) that encourages mode-covering behaviour in the posterior. We derive a VIMCO Estimator for our model in Appendix C.3.

## 4. Experiments

We compared the performance of our VIPR methods with that of Zhang and Matsen IV 2024 (denoted VBPI in this section). We do not compare VIPR to cubeVB (Bouck-

aert, 2024) because cubeVB does not involve maximization and therefore is not directly comparable. However, we do investigate how many tree topologies in the posterior fall outside of the restricted cube space of cubeVB in Appendix B.1. We studied eleven commonly used genetic datasets that are listed in Lakner et al. (2008) denoted DS1 through DS11 (these are the names that are given to these datasets in Lakner et al. 2008).

We also studied a dataset of 72 COVID-19 genomes obtained from GISAID (Global Initiative on Sharing All Influenza Data; Khare et al. 2021). In particular, we obtained COVID-19 RNA sequences that were collected in Canada on January 2, 2025; submitted to GISAID prior to January 20, 2025; contained at least 29,000 sequenced base pairs; and were of the strain JN.1. The 72 COVID-19 genomes studied here are all of the COVID-19 genomes provided by GISAID that satisfied all of these criteria. After obtaining these genomes, we aligned them using multiple sequence alignment in MAFFT using the FFT-NS-1 algorithm (Katoh and Standley, 2013). Finally, we subset the genomes to $M = 3,101$ non-homologous sites (*i.e.*, we omitted all sites that were the same across all 72 taxa). The final datasets ranged from 27 to 72 total taxa $N$ and 378 to 3,101 total sites $M$. See Appendix B.2 for the number of taxa and sites by dataset as well as other summary information.

For all methods considered (BEAST, VBPI and our VIPR methods), we used a Kingman coalescent prior on the phylogenies. We fixed the effective population size at $N_e = 5$ (Kingman, 1982) and assumed the Jukes-Cantor model for mutation (Jukes and Cantor, 1969). These assumptions are described above in Sections 2.2 and 2.3. We also measure the branch lengths in terms of expected mutations per site, which is in line with BEAST and VBPI.

Each run for the experiments on the DS1 to DS11 datasets was executed on a supercomputer node. The runs were allocated 12 hours of wallclock time, 1 CPU, and 16GB of RAM. The supercomputer had a heterogeneous infrastructure involving in which each CPU make and model was Intel v4 Broadwell, Intel Caskade Lake or Skylake, or AMD EPYC 7302. Experiments on the COVID-19 dataset were run with identical conditions to those for DS1 to DS11, but without the 12 hour limit on wallclock time. Instead, they were run for 10,000 iterations (*i.e.*, parameter updates) or 12 hours (whichever took longer).

### 4.1. The BEAST Gold Standard

To approximate the true posterior distribution of each dataset we ran 10 independent MCMC chains using BEAST, each with 10,000,000 iterations. We discarded the first 250,000 iterations as burn-in and thinned to every 1,000-th iteration. This yielded in a total of 97,500 trees that were used as a "gold standard." We estimated ground-truth marginal log-

likelihood values using the stepping-stone estimator (Xie et al., 2010). For each dataset, we ran 100 path steps of 500,000 MCMC iterations and repeated this process ten times to obtain 10 independent estimates of the MLL.

### 4.2. The VBPI Baseline

We compared VIPR to the VBPI algorithm as implemented by Zhang and Matsen IV (2024), which requires MCMC runs to determine likely subsplits (*i.e.*, evolutionary branching events). To provide these runs, we used BEAST to obtain a rooted subsplit support. We ran 10 independent MCMC chains for 1,000,000 iterations, with the first 250,000 discarded as burn-in. We then thinned to every 1,000-th iteration, yielding 7,500 trees for the VBPI subsplit support.

To fit the VBPI baseline, we used the VIMCO gradient estimator with $K$-sample ELBO for $K = 10$ and $K = 20$ (indicated by VBPI10 and VBPI20 in our plots and tables below). Zhang and Matsen IV (2024) use an annealing schedule during optimization, but we omitted the annealing schedule to be consistent with our optimization for VIPR. We used the Adam optimization algorithm implemented in PyTorch with four random restarts and learning rates of 0.003, 0.001, 0.0003, and 0.0001 (Kingma and Ba, 2014; Paszke et al., 2019). We estimated the marginal log-likelihood (MLL) every 100 iterations (*i.e.*, parameter updates) using 500 importance-sampled particles.

Of the 16 runs for each VBPI batch size condition (4 learning rates and 4 random restarts), we retained the run with the highest average MLL in the last 10 estimates of the run. This run (with highest average MLL) was included in our plots and figures. We used the primary subsplit pair (PSP) parameterization of VBPI. Code for these experiments was adapted from https://github.com/zcrabbit/vbpi-torch/tree/main/rooted. See Zhang and Matsen IV (2024), Section 6 for more implementation details.

### 4.3. The VIPR Methods

For our VIPR methods, we set the variational distributions $q_\phi^{\{u,v\}}$ to be log-normal, so the variational parameters $\phi$ were the means and standard deviations corresponding to the logarithm of the entries $\log(t^{\{u,v\}})$ of the matrix $\mathbf{T}$. After running BEAST, we plotted histograms of pairwise coalescent times across sampled trees for all datasets. In most cases these histograms looked approximately log-normal, motivating our choice of log-normal distributions (see https://github.com/EvanSidrow/VIPR/tree/main/supmat/hists for the histograms). We also simulated data and ran posterior predictive checks in Appendix B.3 for model checking.

To initialize the parameters $\phi$, we computed the empirical

distribution of coalescent times between taxa $\{u, v\}$ from the short MCMC runs used to establish the support for the VBPI baseline. We then set the initial mean and standard deviation of $q_\phi^{\{u,v\}}$ to be the mean and standard deviation of the empirical distribution.

We experimented with three gradient estimation techniques. We estimated $\nabla_\phi L(\phi)$ using (1) the LOOR estimator and (2) the reparameterization trick, both with batch sizes of 10 samples. We also estimated $\nabla_\phi L_K(\phi)$ using the VIMCO estimator with a batch size of $K = 10$.

For each gradient estimation technique, we used the Adam optimizer in PyTorch with ten random restarts and learning rates of 0.001, 0.003, 0.01, and 0.03. We recorded the estimated MLL every 10 iterations (*i.e.*, parameter updates) with 50 Monte Carlo samples. Of the 40 runs (4 learning rates and 10 random restarts), we retained the run with the highest average MLL in the last 10 estimates of the run. As for VBPI, the retained run (for each dataset and technique) is reported in the plots and figures below.

### 4.4. Simulated Datasets

To explore the runtime and asymptotic complexity of the methods as a function of the number of taxa, we created a series of simulated datasets. Through this simulation, we could keep the number of sites and the underlying tree distribution and substitution model all constant. We simulated seven datasets using the *ms* software (Hudson, 2002) with 1,000 sites, an infinite sites model (Kimura, 1969), a neutral model of evolution, and the Kingman coalescent. These assumptions imply the software command 'ms $\langle N \rangle$ 1 -T -s 1000.' Here $\langle N \rangle$ indicates the number of taxa, which we varied in the set $\{8, 16, 32, 64, 128, 256, 512\}$. We refer to these datasets as the MS datasets.

### 5. Results

Tables 1 and 2 show the estimated MLLs and ELBOs for our variational inference experiments after 12 hours of compute time. Results for VBPI with $K = 20$ (VBPI20) are in Appendix B.4. The stepping-stone algorithm is not a variational method, and so it has no entry in Table 2.

The MLLs in Table 1 are reported by the gap between the MLL of the gold standard (BEAST/stepping stone run) and the method's MLL (the difference between the MLLs). Methods with smaller gaps are therefore closer to the gold standard, and the method with the highest MLL is bolded. Note that some VI methods surpass the gold standard, likely due to Monte Carlo error.

VBPI tends to slightly outperform our VIPR methods in terms of MLL, but all methods are comparable in terms ELBO (with our methods outperforming VBPI on exactly

half of the datasets). This is likely because VBPI targets a multi-sample ELBO for optimization, which produces mode-covering behaviour. In contrast, our VIPR methods target the single-sample ELBO.

*Table 1.* **Gap between gold standard and estimated marginal log-likelihoods for variational inference methods (in nats).** *Marginal log-likelihoods for VI methods were estimated using importance sampling with 1,000 random samples from each variational distribution. Values indicate difference between gold standard MLLs and each method's MLLs. Gold standard MLLs (indicated by the* BEAST *column) are derived from 10 independent chains of the stepping-stone algorithm in BEAST. Datasets (*DATA *column) DS1 to DS11 are from Lakner et al. (2008). Dataset COV is the COVID-19 dataset obtained from GISAID. VI methods are specified by columns: Variational Bayesian Phylogenetic Inference with K-sample ELBO, $K = 10$ (*VBPI10*; Zhang and Matsen IV 2024); VIPR using the leave-one-out REINFORCE estimator (*LOOR*); VIPR using the reparameterization trick (*REP*); VIPR using the Variational Inference for Monte Carlo Objectives estimator with $K = 10$ (*VIMCO*). Results for VBPI20 and standard errors are in Appendix B.4.*

| DATA | BEAST | VBPI10 | LOOR | REP | VIMCO |
|------|-------|--------|------|-----|-------|
| DS1 | −7154.26 | **−0.53** | −2.29 | −1.83 | −0.95 |
| DS2 | −26566.42 | **0.16** | −0.76 | −0.14 | −0.37 |
| DS3 | −33787.62 | **−0.44** | −3.66 | −1.91 | −2.63 |
| DS4 | −13506.05 | **0.03** | −2.48 | −0.47 | −1.73 |
| DS5 | −8271.26 | −1.70 | −0.29 | −4.01 | **0.94** |
| DS6 | −6745.31 | **−0.76** | −3.96 | −3.26 | −2.72 |
| DS7 | −37323.88 | **0.27** | −2.73 | −2.82 | −10.42 |
| DS8 | −8650.20 | **−0.82** | −3.28 | −4.95 | −2.88 |
| DS9 | −4072.66 | −5.32 | **−3.12** | −5.79 | −7.60 |
| DS10 | −10102.65 | **−0.88** | −5.38 | −3.98 | −6.82 |
| DS11 | −6272.57 | −18.79 | **−6.79** | −7.31 | −9.62 |
| COV | −7861.61 | **−39.1** | −611 | −374 | −214 |

Figure 2 shows the trace of estimated log-likelihood versus iteration number for all VI methods on DS1 and on the COVID-19 dataset. See Appendix B.4 for results on DS2-11. These figures also display empirical distributions of tree metrics for each VI method's learned variational distribution in addition to the BEAST gold standard run (plotted with matplotlib's *kde* function with default parameters; Hunter 2007). We removed 2 of the 1,000 trees sampled from VBPI20 for the COVID-19 experiment because they had extremely low log-likelihoods ($< −90,000$), resulting in flat densities.

VIPR tended to underestimate the variance of tree length compared to BEAST, while VBPI tended to overestimate. In addition, the reinforce and reparameterization gradient estimates result in variational distributions with higher tree log-likelihoods on average, while VBPI and our VIPR with

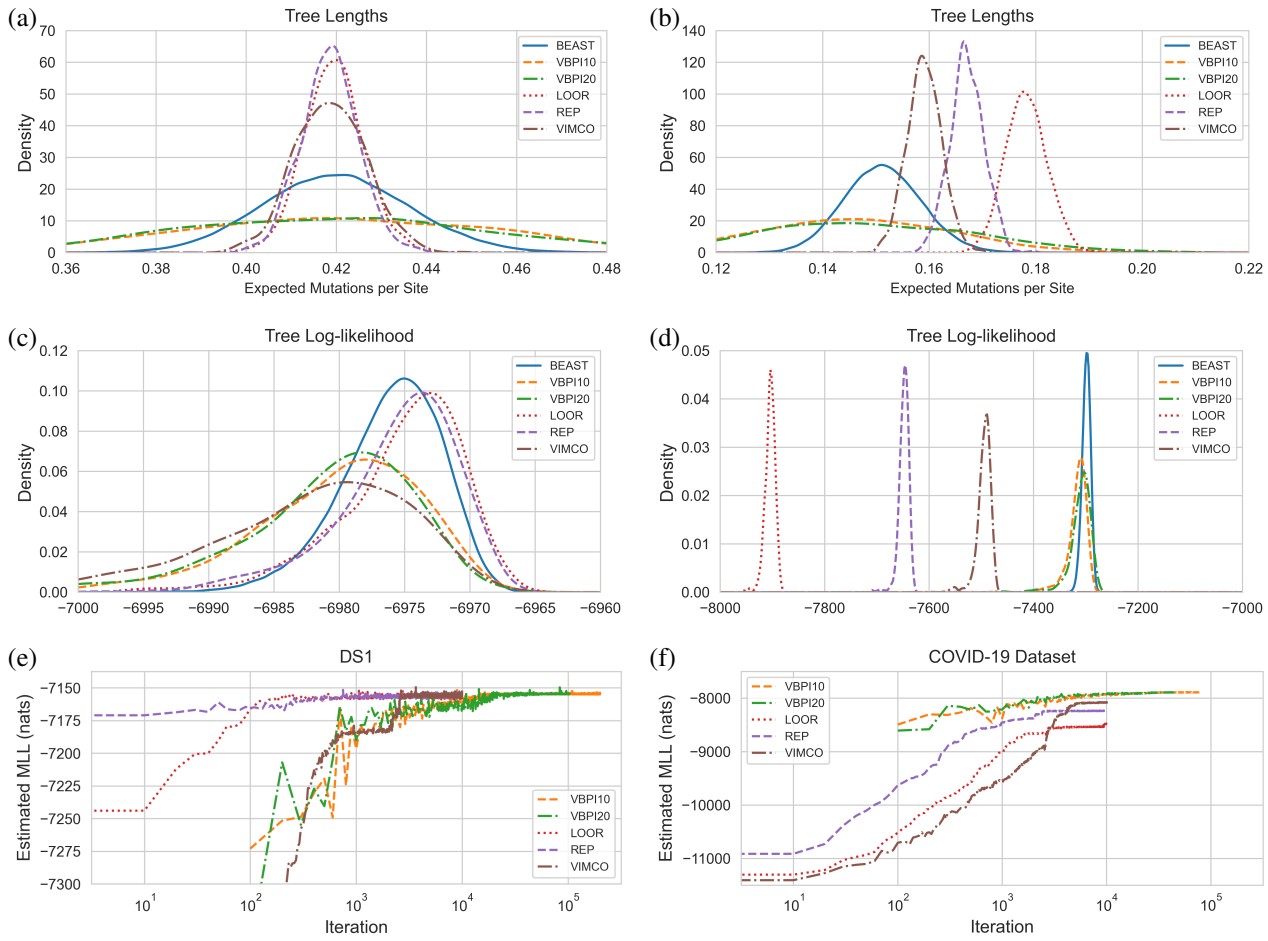

Figure 2. **Variational inference results for DS1 (left) and the COVID-19 dataset (right).** (a–b) *Density estimation for tree lengths.* (c–d) *Density estimation for tree log-likelihoods. Estimates are formed from 1,000 samples from the variational posterior of each VI method and 97,500 samples from the BEAST gold standard.* (e–f) *Trace plots of estimated marginal log-likelihood vs. iteration number. The number of importance samples used to estimate the marginal log-likelihood was 500 for VBPI and 50 for VIPR.*

VIMCO tended to produce trees with more variable log-likelihood values. Again, this is likely because the multisample ELBO results in mode-covering behaviour.

VIPR converged quickly on DS1 because its parameters were initialized in a region of high ELBO, while VIPR converged slower for the COVID-19 dataset since its parameters were initialized in a region of low ELBO. The optimization may have been caught in relatively flat regions of the parameter space, highlighting the need for intelligent parameter initializations or annealing schedules.

### 5.1. Computational Complexity

To compare the time complexity of our algorithm against VBPI, we considered the MS datasets (described in Section 4.4) with 1,000 sites and between 8 and 512 taxa. We ran each method for either 5 minutes or 1,000 iterations (whichever came first) and plotted the wall clock time per 1,000 iterations. These experiments were run on a 2019

Macbook Pro with 16GB of RAM and a 2.6 GHz 6-core Intel i7 CPU. The wallclock time in seconds for each method versus the number of taxa is shown in Figure 3. This is also plotted in terms of log ratios in Appendix B.5 (with slope indicating complexity). Our method is approximately twice as slow as VBPI per iteration for 8 taxa, but it scales better and outperforms VBPI for 512 taxa. Even though evaluating the variational density of VIPR takes $\mathcal{O}(N^2)$ time, VIPR has an empirical time complexity of roughly $\mathcal{O}(N)$, indicating that the primary bottleneck is calculating the likelihood, which takes $\mathcal{O}(NM)$ time. We also demonstrated that the number of parameters for VBPI grows super-linearly with the number of taxa $N$ (see Appendix B.5), so the asymptotic computational complexity of VBPI may also be super-linear.

For SBNs, as the number of taxa grows, the number of parameters grows with the number of trees in the SBN. There is no closed form for this number—it depends on the MCMC and the posterior concentration. In Table 3 we show

Table 2. ***Estimated evidence lower bounds for variational inference methods (in nats).*** *ELBOs were estimated using importance sampling on 1,000 random samples from each variational distribution. Our VIPR methods beat the VBPI baseline on half of the datasets. Dataset names, method acronyms, and conditions are the same as those described in the caption for Table 1.*

| DATA | VBPI10 | LOOR | REP | VIMCO |
|------|--------|------|-----|-------|
| DS1 | **−7157.99** | −7159.56 | −7159.54 | −7161.60 |
| DS2 | −26573.03 | −26569.56 | **−26569.50** | −26570.74 |
| DS3 | **−33793.96** | −33794.96 | −33794.77 | −33796.53 |
| DS4 | −13541.39 | **−13512.54** | −13512.60 | −13513.41 |
| DS5 | −8281.03 | **−8279.93** | −8280.35 | −8282.03 |
| DS6 | **−6751.77** | −6754.36 | −6755.29 | −6756.10 |
| DS7 | **−37331.12** | −37333.36 | −37332.04 | −37352.10 |
| DS8 | **−8657.78** | −8662.26 | −8661.88 | −8664.54 |
| DS9 | −4088.64 | **−4085.61** | −4087.25 | −4090.52 |
| DS10 | **−10111.81** | −10114.76 | −10115.16 | −10119.70 |
| DS11 | −6329.37 | **−6289.60** | −6289.70 | −6294.31 |
| COV | −8100.96 | −8489.82 | −8244.41 | **−8087.43** |

the number of parameters in the SBNs for VBPI on each of the seven MS datasets. We compare to VIPR, showing quadratic behaviour for VIPR and that VBPI uses more parameters when the number of taxa is greater than 128.

## 6. Discussion

In this work, we introduce a new variational family over ultrametric, time-measured phylogenies that models the coalescent time between each pair of taxa. The family is formed by deriving a closed-form expression for the marginal distribution on phylogenies induced by single linkage clustering on a distance matrix. Methods using this variational family require only $\mathcal{O}(\binom{N}{2})$ parameters in total, and each parameter has an intuitive interpretation as a description of the distribution on pairwise coalescents. For example, in this work we place independent log-normal distributions on the entries of the distance matrix, yielding $2\binom{N}{2}$ parameters (one mean and one standard deviation for each pair of taxa). VIPR is unique in that it does not *require* aspects of MCMC runs in its iterations in order to make inference computationally tractable. In contrast, Zhang and Matsen IV (2024) used MCMC to fix the support of the trees described by their variational family. (Note that we also used short MCMC runs to initialize our parameters.)

Our methods may be further developed in many ways—for example, by moving from the log-normal distribution on pairwise coalescent times to mixture distributions Molén et al. (2024), or by using normalizing flows similar to Zhang (2020). We could also directly enforce sparsity in the prior

Table 3. ***Number of tree structure parameters versus number of taxa*** (**NTAXA**) ***on simulated data with 1,000 sites.***

| NTAXA | VBPI | VIPR |
|-------|------|------|
| 8 | 4 | 56 |
| 16 | 44 | 240 |
| 32 | 55 | 992 |
| 64 | 3,826 | 4,032 |
| 128 | 29,939 | 16,256 |
| 256 | 127,217 | 65,280 |
| 512 | 319,533 | 261,632 |

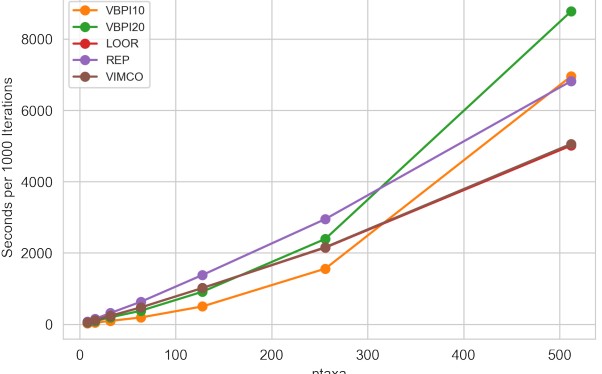

Figure 3. ***Seconds per 1,000 iterations vs. number of taxa.*** *Each VI method was run for 1,000 iterations or 5 minutes (whichever took less) on simulated datasets.*

by fixing the distribution on the time to coalescence of taxa $u$ and $v$ that are far away in genetic space at infinity, thus reducing the number of parameters to learn. Expanding the variational family to include conditional parameters may also improve performance: if taxa $u$ and $v$ coalesce first, we may define a new parameter $\phi^{(\{u,v\},w)}$ describing the coalesce between a clade containing $\{u,v\}$ and another taxon $w$. Further, fast and accurate parameter initializations and well-tuned annealing are essential for top performance in variational Bayesian phylogenetics. Our experiments may be improved by using an annealing schedule similar to Zhang and Matsen IV (2024) to prevent convergence to local maxima.

We have focused on the difficult task of inferring tree topology and branch lengths. Inference for aspects such as a relaxed clock (see Douglas et al., 2021) and effective populations size can also be done starting from this new variational family. Our method is thus a promising foundation on which more intricate variational families can be built.

## Acknowledgements

This research was enabled in part by support provided by the Digital Research Alliance of Canada. We gratefully acknowledge all data contributors, *i.e.*, the authors and their originating laboratories responsible for obtaining the specimens, and their submitting laboratories for generating the genetic sequence and metadata and sharing via the GISAID Initiative, on which this research is based. We also thank the anonymous reviewers for their helpful comments.

## Impact Statement

This paper presents work that advances the field of machine learning. There are many potential societal consequences of machine learning (see for example Metz 2023). However, applications of our methods may also advance basic science in diverse fields that use phylogenetics such as epidemiology, linguistics and ecology.

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

## A. Proof for Proposition 1

We provide a proof by induction for the derivation of Equation (7). Consider the coalescent events in Algorithm 2. For $1 \le K \le N-1$, let $\mathbf{t}_{1:K}$ be the times of the first $K$ coalescent events ($\mathbf{t}_{1:K} = \{t_n\}_{n=1}^K$). Let $\tau_{1:K}$ be the bipartitions of the first $K$ coalescent events ($\tau_{1:K} = \{\{W_n, Z_n\}\}_{n=1}^K$). Let $q_{\phi,K}(\tau_K, \mathbf{t}_K)$ be the probability density function of the marginal distribution on the times and the bipartitions of the first $K$ coalescent events. Let $\mathcal{S}_n$ be the set $\{\{w, z\} : w \in W_n, z \in Z_n\}$ (here $\{W_n, Z_n\}$ is the $n$-th bipartition). Note that $\mathcal{S}_n$ is the set of all unordered pairs of taxa that have not coalesced before $t_n$ and that coalesce at $t_n$. Let $\mathcal{S}_{1:K}$ be $\bigcup_{n=1}^K \mathcal{S}_n$ (*i.e.*, $\mathcal{S}_{1:K}$ is the set of all unordered pairs of taxa that coalesce by time $t_n$). By the definition of $\mathcal{S}_n$, a sum $\sum_{w \in W_n, z \in Z_n} \cdot$ is equal to the same sum indexed by $\{w, z\} \in \mathcal{S}_n$. (And the same is true of products.) Our induction hypothesis for $1 \le K \le N-1$ is as follows:

$$q_{\phi,K}(\tau_{1:K}, \mathbf{t}_{1:K}) = \prod_{n=1}^K \left( \left( \sum_{\{w,z\} \in \mathcal{S}_n} \frac{q_\phi^{\{w,z\}}(t_n)}{Q_\phi^{\{w,z\}}(t_n)} \right) \prod_{\{w,z\} \in \mathcal{S}_n} Q_\phi^{\{w,z\}}(t_n) \right) \prod_{\{w,z\} \notin \mathcal{S}_{1:K}} Q_\phi^{\{w,z\}}(t_K). \tag{10}$$

In Equation (10), the product over $\{w, z\} \notin \mathcal{S}_{1:K}$ is outside of the product over $n = 1, \ldots, K$. (Throughout this derivation, if a product has more than one factor in its operand, they are all enclosed by the pair of brackets appearing immediately after the product sign.) Consider the base case of the induction where $K = 1$. There exists an unordered pair of taxa $\{w^*, z^*\}$ such that $\{W_1, Z_1\} = \{\{w^*\}, \{z^*\}\}$. The probability density $q_{\phi,1}(\tau_{1:1}, \mathbf{t}_{1:1})$ is the density of the event that taxa $w^*$ and $z^*$ coalesce at time $t_1$ (this density is $q^{\{w^*, z^*\}}(t_1)$) times the probability that all other taxa coalesce after time $t_1$ (as $\{W_1, Z_1\}$ is the first bipartition). Therefore, we have:

$$q_{\phi,1}(\tau_{1:1}, \mathbf{t}_{1:1}) = q_\phi^{\{w^*, z^*\}}(t_1) \prod_{\{w,z\} \neq \{w^*, z^*\}} Q_\phi^{\{w,z\}}(t_1) \tag{11}$$

$$= \frac{q_\phi^{\{w^*, z^*\}}(t_1)}{Q_\phi^{\{w^*, z^*\}}(t_1)} Q_\phi^{\{w^*, z^*\}}(t_1) \prod_{\{w,z\} \notin \mathcal{S}_1} Q_\phi^{\{w,z\}}(t_1) \tag{12}$$

$$= \left( \sum_{\{w,z\} \in \mathcal{S}_1} \frac{q_\phi^{\{w,z\}}(t_1)}{Q_\phi^{\{w,z\}}(t_1)} \right) \left( \prod_{\{w,z\} \in \mathcal{S}_1} Q_\phi^{\{w,z\}}(t_1) \right) \prod_{\{w,z\} \notin \mathcal{S}_{1:1}} Q_\phi^{\{w,z\}}(t_1). \tag{13}$$

Here in Equation (11) we use the mutual independence of $t^{(\cdot, \cdot)}$ to split the joint probability of all taxa coalescing after $t_1$, other than $\{w^*, z^*\}$, into a product. Thus, the base case (where $K = 1$) is established. Assume that the induction hypothesis in Equation (10) holds for a given $K - 1$ (here $1 \le K - 1 < N - 1$). Consider the conditional probability density function $q_{\phi,K}(\{W_K, Z_K\}, t_K \mid \tau_{1:K-1}, \mathbf{t}_{1:K-1})$. When we condition on $\tau_{1:K-1}, \mathbf{t}_{1:K-1}$, the $K$-th coalescent event with bipartition $\{W_K, Z_K\}$ occurs at time $t_K$ if and only if the following hold:

1. There exists an unordered pair of taxa $\{w^*, z^*\} \in \mathcal{S}_K$ such that $t^{\{w^*, z^*\}} = t_K$. (We are conditioning on the event that taxa $w^*$ and $z^*$ have not coalesced before time $t_{K-1}$.) The conditional probability density of the event that $w^*$ and $z^*$ coalesce at time $t_K$ is thus $q_\phi^{\{w^*, z^*\}}(t_K)/Q_\phi^{\{w^*, z^*\}}(t_{K-1})$.

2. All *other* taxa pairs that have not coalesced by time $t_{K-1}$ coalesce after time $t_K$. (As the $q_\phi^{\{\cdot, \cdot\}}$'s are continuously differentiable, they are continuous and so $\{w^*, z^*\}$ is unique almost surely.) Note that we are conditioning on the event that all taxa pairs $\{w, z\} \notin \mathcal{S}_{1:K-1}$ have not coalesced before time $t_{K-1}$. The conditional probability density of the event that $w^*$ and $z^*$ have not coalesced by time $t_K$ is thus $Q_\phi^{\{w^*, z^*\}}(t_K)/Q_\phi^{\{w^*, z^*\}}(t_{K-1})$.

The conditional probability density is thus:

$$q_{\phi,K}(\{W_K, Z_K\}, t_K \mid \tau_{1:K-1}, \mathbf{t}_{1:K-1}) \tag{14}$$

$$= \sum_{\substack{\{w^*,z^*\} \in \mathcal{S}_K}} \left( \frac{q_\phi^{\{w^*,z^*\}}(t_K)}{Q_\phi^{\{w^*,z^*\}}(t_{K-1})} \prod_{\substack{\{w,z\} \notin \mathcal{S}_{1:K-1} \\ \{w,z\} \neq \{w^*,z^*\}}} \frac{Q_\phi^{\{w,z\}}(t_K)}{Q_\phi^{\{w,z\}}(t_{K-1})} \right) \tag{15}$$

$$= \left( \sum_{\{w,z\} \in S_K} \frac{q_\phi^{\{w,z\}}(t_K)}{Q_\phi^{\{w,z\}}(t_K)} \right) \prod_{\{w,z\} \notin \mathcal{S}_{1:K-1}} \frac{Q_\phi^{\{w,z\}}(t_K)}{Q_\phi^{\{w,z\}}(t_{K-1})} \tag{16}$$

$$= \left( \sum_{\{w,z\} \in S_K} \frac{q_\phi^{\{w,z\}}(t_K)}{Q_\phi^{\{w,z\}}(t_K)} \right) \left( \prod_{\{w,z\} \in S_K} \frac{Q_\phi^{\{w,z\}}(t_K)}{Q_\phi^{\{w,z\}}(t_{K-1})} \right) \prod_{\{w,z\} \notin \mathcal{S}_{1:K}} \frac{Q_\phi^{\{w,z\}}(t_K)}{Q_\phi^{\{w,z\}}(t_{K-1})}. \tag{17}$$

Note that we drop the stars on the taxa $w$ and $z$ after Equation (15) because the indices no longer need to be distinguished once the sum is isolated. Also, in Equation (15) we use the mutual independence of $t^{(\cdot,\cdot)}$ to form the product. Multiplying this conditional probability with the induction hypothesis Equation (10) for $K-1$ yields the total probability density:

$$q_{\phi,K}(\tau_{1:K}, \mathbf{t}_{1:K}) = q_{\phi,K}(\{W_K, Z_K\}, t_K \mid \tau_{1:K-1}, \mathbf{t}_{1:K-1}) \cdot q_{\phi,K-1}(\tau_{1:K-1}, \mathbf{t}_{1:K-1}) \tag{18}$$

$$= \left( \sum_{\{w,z\} \in \mathcal{S}_K} \frac{q_\phi^{\{w,z\}}(t_K)}{Q_\phi^{\{w,z\}}(t_K)} \right) \left( \prod_{\{w,z\} \in \mathcal{S}_K} \frac{Q_\phi^{\{w,z\}}(t_K)}{Q_\phi^{\{w,z\}}(t_{K-1})} \right) \left( \prod_{\{w,z\} \notin \mathcal{S}_{1:K}} \frac{Q_\phi^{\{w,z\}}(t_K)}{Q_\phi^{\{w,z\}}(t_{K-1})} \right) \tag{19}$$

$$\cdot \prod_{n=1}^{K-1} \left( \left( \sum_{\{w,z\} \in \mathcal{S}_n} \frac{q_\phi^{\{w,z\}}(t_n)}{Q_\phi^{\{w,z\}}(t_n)} \right) \prod_{\{w,z\} \in S_n} Q_\phi^{\{w,z\}}(t_n) \right) \prod_{\{w,z\} \notin \mathcal{S}_{1:K-1}} Q_\phi^{\{w,z\}}(t_{K-1}). \tag{20}$$

We then move the first component of Line (19) into the first component of Line (20), and we move the numerator of the second component of Line (19) into the second component of Line (20). These rearrangements yield the following:

$$q_{\phi,K}(\tau_{1:K}, \mathbf{t}_{1:K}) = \left( \prod_{\{w,z\} \in \mathcal{S}_K} \frac{1}{Q_\phi^{\{w,z\}}(t_{K-1})} \right) \left( \prod_{\{w,z\} \notin \mathcal{S}_{1:K}} \frac{Q_\phi^{\{w,z\}}(t_K)}{Q_\phi^{\{w,z\}}(t_{K-1})} \right)$$

$$\cdot \prod_{n=1}^{K} \left( \left( \sum_{\{w,z\} \in \mathcal{S}_n} \frac{q_\phi^{\{w,z\}}(t_n)}{Q_\phi^{\{w,z\}}(t_n)} \right) \prod_{\{w,z\} \in \mathcal{S}_n} Q_\phi^{\{w,z\}}(t_n) \right) \prod_{\{w,z\} \notin \mathcal{S}_{1:K-1}} Q_\phi^{\{w,z\}}(t_{K-1}). \tag{21}$$

For the final display, in Equation (22) we split the numerator and denominator of $Q_\phi^{\{w,z\}}(t_K)/Q_\phi^{\{w,z\}}(t_{K-1})$ into separate products. And in Equation (23) we cancel the $Q_\phi^{\{w,z\}}(t_{K-1})$ factors involving $\{w,z\} \in \mathcal{S}_K$. And in Equation (24) we cancel the $Q_\phi^{\{w,z\}}(t_{K-1})$ factors involving $\{w,z\} \notin \mathcal{S}_{1:K-1}$.

$$q_{\phi,K}(\tau_{1:K}, \mathbf{t}_{1:K}) = \left( \prod_{\{w,z\} \in \mathcal{S}_K} \frac{1}{Q_\phi^{\{w,z\}}(t_{K-1})} \right) \left( \prod_{\{w,z\} \notin \mathcal{S}_{1:K}} \frac{1}{Q_\phi^{\{w,z\}}(t_{K-1})} \right) \left( \prod_{\{w,z\} \notin \mathcal{S}_{1:K}} Q_\phi^{\{w,z\}}(t_K) \right)$$
$$\cdot \prod_{n=1}^{K} \left( \left( \sum_{\{w,z\} \in \mathcal{S}_n} \frac{q_\phi^{\{w,z\}}(t_n)}{Q_\phi^{\{w,z\}}(t_n)} \right) \prod_{\{w,z\} \in \mathcal{S}_n} Q_\phi^{\{w,z\}}(t_n) \right) \prod_{\{w,z\} \notin \mathcal{S}_{1:K-1}} Q_\phi^{\{w,z\}}(t_{K-1}) \tag{22}$$

$$= \left( \prod_{\{w,z\} \notin \mathcal{S}_{1:K-1}} \frac{1}{Q_\phi^{\{w,z\}}(t_{K-1})} \right) \left( \prod_{\{w,z\} \notin \mathcal{S}_{1:K}} Q_\phi^{\{w,z\}}(t_K) \right)$$
$$\cdot \prod_{n=1}^{K} \left( \left( \sum_{\{w,z\} \in \mathcal{S}_n} \frac{q_\phi^{\{w,z\}}(t_n)}{Q_\phi^{\{w,z\}}(t_n)} \right) \prod_{\{w,z\} \in \mathcal{S}_n} Q_\phi^{\{w,z\}}(t_n) \right) \prod_{\{w,z\} \notin \mathcal{S}_{1:K-1}} Q_\phi^{\{w,z\}}(t_{K-1}) \tag{23}$$

$$= \prod_{n=1}^{K} \left( \left( \sum_{\{w,z\} \in \mathcal{S}_n} \frac{q_\phi^{\{w,z\}}(t_n)}{Q_\phi^{\{w,z\}}(t_n)} \right) \prod_{\{w,z\} \in \mathcal{S}_n} Q_\phi^{\{w,z\}}(t_n) \right) \prod_{\{w,z\} \notin \mathcal{S}_{1:K}} Q_\phi^{\{w,z\}}(t_K). \tag{24}$$

Thus, the inductive step is established and Equation (10) holds for all $1 \le K \le N - 1$. To complete the derivation, note that $q_{\phi,N-1} = q_\phi$; and $\mathcal{S}_{1:N-1} = \bigcup_{n=1}^{N-1} \mathcal{S}_n = \{\{w,z\} : w, z \in \mathcal{X}\}$ (*all* taxa coalesce after $N - 1$ coalescent events); and indices over $w \in W_n, z \in Z_n$ are equivalent to indices over $\{w, z\} \in S_n$. Thus, for $K = N - 1$ the last term of Equation (24) is an empty product yielding the desired result:

$$q_\phi(\tau, \mathbf{t}) = q_{\phi,N-1}(\tau_{1:N-1}, \mathbf{t}_{1:N-1}) \tag{25}$$
$$= \prod_{n=1}^{N-1} \left( \left( \sum_{\{w,z\} \in \mathcal{S}_n} \frac{q_\phi^{\{w,z\}}(t_n)}{Q_\phi^{\{w,z\}}(t_n)} \right) \prod_{\{w,z\} \in \mathcal{S}_n} Q_\phi^{\{w,z\}}(t_n) \right) \tag{26}$$
$$= \prod_{n=1}^{N-1} \left( \left( \sum_{\substack{w \in W_n \\ z \in Z_n}} \frac{q_\phi^{\{w,z\}}(t_n)}{Q_\phi^{\{w,z\}}(t_n)} \right) \prod_{\substack{w \in W_n \\ z \in Z_n}} Q_\phi^{\{w,z\}}(t_n) \right). \tag{27}$$

# B. Additional Results

## B.1. Coverage of cubeVB

Our main comparison is between our method VIPR and the method of Zhang and Matsen IV (2024). Both of these methods are based on optimization. However, another variational inference method for ultrametric phylogenetic trees is described by Bouckaert (2024). We do not compare to this method in terms of likelihood estimation because it is not optimization-based. Nonetheless, to compare VIPR with Bouckaert (2024), we constructed the maximum clade credibility (MCC) tree using our gold standard BEAST run, selected an ordering of taxa at random using the MCC tree, and calculated the percentage of tree topologies from the BEAST gold-standard that are within the "cube space" implied by the ordering. This process estimates the percentage of the posterior that is impossible to reach using the restricted tree space from Bouckaert (2024):

*Table 4.* Percentage of trees sampled from BEAST that are outside of cube space as defined by Bouckaert (2024). Percentages are listed for each dataset considered in our likelihood experiments. In many cases, the coverage of cube space is extremely small. Dataset sizes are provided below in Table 5.

| DATA | % OUTSIDE CUBE SPACE |
|------|----------------------|
| DS1  | 29.2 |
| DS2  | 15.2 |
| DS3  | 76.8 |
| DS4  | 79.7 |
| DS5  | 98.0 |
| DS6  | 94.7 |
| DS7  | 69.9 |
| DS8  | 42.7 |
| DS9  | 99.9 |
| DS10 | 84.6 |
| DS11 | 99.9 |
| COV  | 99.9 |

Some datasets correspond to posteriors where 99.9% of sampled trees lie outside of cube space. Although these results are striking, the discussion of Bouckaert (2024) mentions that CubeVB may struggle on high-entropy posteriors.

## B.2. Dataset Characteristics

To understand the characteristics of each dataset, we calculated pairwise Hamming distances between each taxa for each dataset (dropping sites with missingness). These dataset features may be cross referenced with the likelihood estimation in Section 5 to better understand relative model performance.

*Table 5.* Number of taxa $N$, number of sites $M$, and the average Hamming distance between sites for each dataset. Values in parentheses indicate standard deviations.

| DATA | $N$ | $M$ | HAMMING DISTANCE / $M$ |
|------|-----|-----|------------------------|
| DS1  | 27  | 1949 | .040(.017) |
| DS2  | 29  | 2520 | .214(.057) |
| DS3  | 36  | 1812 | .230(.051) |
| DS4  | 41  | 1137 | .138(.055) |
| DS5  | 50  | 378  | .192(.041) |
| DS6  | 50  | 1133 | .056(.029) |
| DS7  | 59  | 1824 | .203(.069) |
| DS8  | 64  | 1008 | .082(.031) |
| DS9  | 67  | 955  | .025(.014) |
| DS10 | 67  | 1098 | .070(.026) |
| DS11 | 71  | 1082 | .082(.053) |
| COV  | 72  | 3101 | .008(.003) |

## B.3. Posterior Predictive Checks

In this Appendix, we run simple posterior predictive checks using 10 trees simulated from a Kingman coalescent model with $N = 8$ taxa and an effective population size of $N_e = 1.0$. For each tree, we simulated genomes using the Jukes-Cantor model of evolution with $M = 1,000$ sites. We then ran VIPR using the leave-one-out reinforce estimator for $10,000$ iterations and the same optimization parameters as the primary experiments. Figure 4 displays histograms of sampling $10,000$ trees from the resulting variational distributions. Our method tends to underestimate tree length (the sum of all branch lengths) for trees of length $8$ or greater. This is likely because the posterior distribution for very long branch lengths is approximately exponential, but we use log-normal distributions for pair-wise distances in VIPR. Besides that notable exception, the true tree length and log-likelihood fit comfortably within the posterior distribution estimated by VIPR.

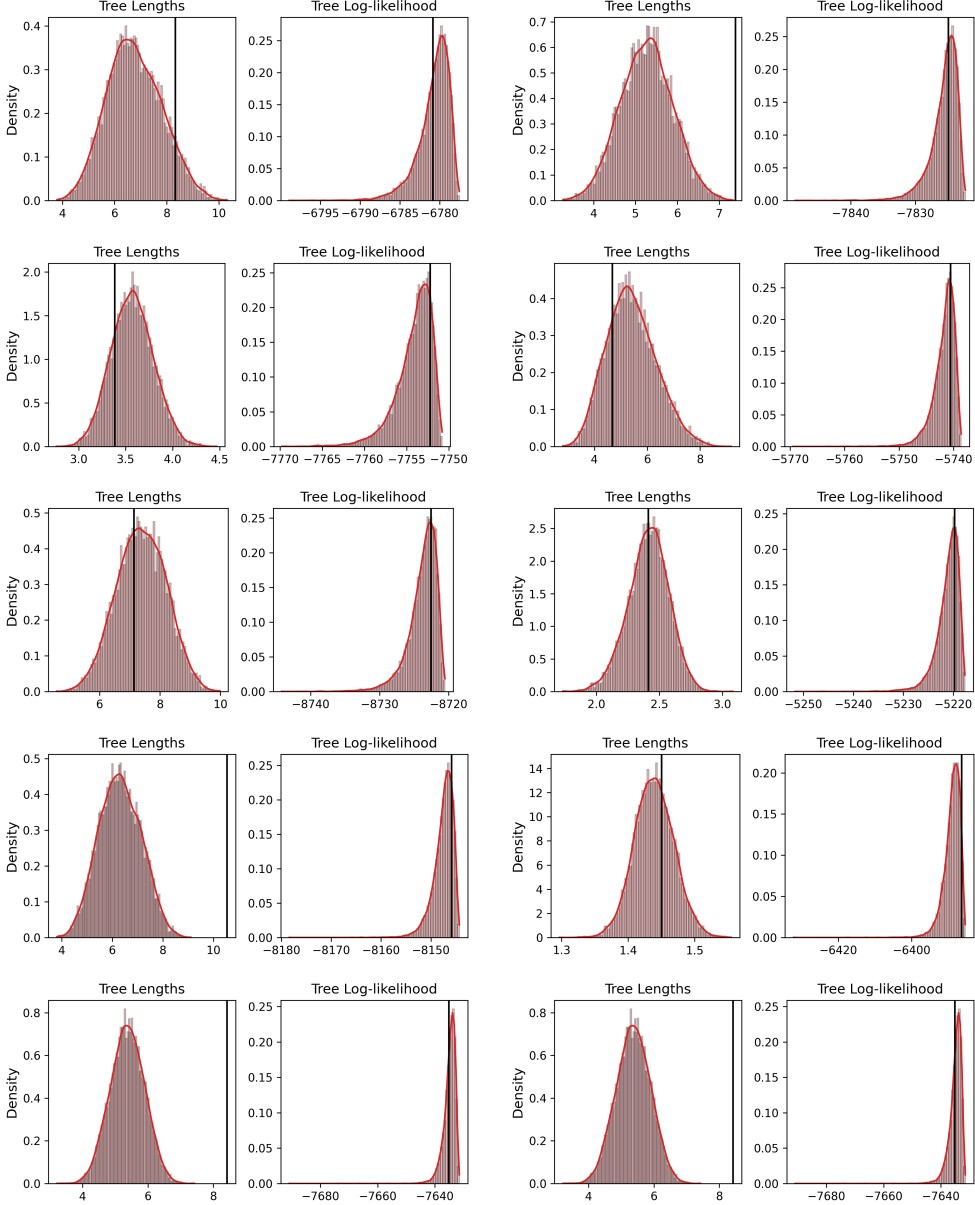

Figure 4. *Posterior predictive checks for VIPR.* Checks include tree lengths (the sum of all branch lengths) and log-likelihoods. True trees were generated using $N = 8$ taxa and a Kingman coalescent with $N_e = 1.0$. Genomes were sampled using a Juke-Cantor model of evolution and $M = 1,000$ sites. Results are shown for $10,000$ samples from the variational posterior (histograms) and the true tree used to generate the genomes (vertical black lines).

## B.4. Marginal Log Likelihood and ELBO Values

In this Appendix, we include more complete versions of Tables 1 and 2 from the main text. In particular, Tables 6 and 7 below include the number of taxa and sites for each dataset, results for VBPI with a batch size of $K = 20$ (VBPI20), and estimated standard errors using 100 bootstrapped samples. Further, Figure 5 below shows trace plots of marginal log-likelihood versus iteration number for all 12 datasets.

*Table 6.* **Gap between gold standard and estimated marginal log-likelihoods for variational inference methods (in nats)**. *Marginal log-likelihoods for VI methods were estimated using importance sampling with 1,000 random samples from each variational distribution. Values indicate differences between gold standard MLLs and each method's MLLs. Gold standard MLLs (indicated by the* BEAST *column) are derived from 10 independent chains of the stepping-stone algorithm in BEAST. Datasets (*DATA *column) DS1 to DS11 are from* Lakner et al. (2008). *Dataset COV is the COVID-19 dataset obtained from GISAID. VI methods are specified by columns: Variational Bayesian Phylogenetic Inference with $K$-sample ELBO, $K = 10$ (*VBPI10*; Zhang and Matsen IV 2024); Variational Bayesian Phylogenetic Inference with $K$-sample ELBO, $K = 20$ (*VBPI20*; Zhang and Matsen IV 2024); VIPR using the leave-one-out REINFORCE estimator (*LOOR*); VIPR using the reparameterization trick (*REP*); VIPR using the Variational Inference for Monte Carlo Objectives estimator with $K = 10$ (*VIMCO*). Standard errors were estimated using 100 bootstrapped samples and are shown in parentheses.*

| DATA | $(N, M)$ | BEAST | VBPI10 | VBPI20 | LOOR | REP | VIMCO |
|------|----------|-------|--------|--------|------|-----|-------|
| DS1 | $(27, 1949)$ | $-7154.26(0.19)$ | $-0.53(0.09)$ | $\mathbf{0.36(0.13})$ | $-2.29(0.15)$ | $-1.83(0.21)$ | $-0.95(0.46)$ |
| DS2 | $(29, 2520)$ | $-26566.42(0.26)$ | $\mathbf{0.16(0.24})$ | $0.01(0.20)$ | $-0.76(0.14)$ | $-0.14(0.43)$ | $-0.37(0.29)$ |
| DS3 | $(36, 1812)$ | $-33787.62(0.36)$ | $-0.44(0.12)$ | $\mathbf{-0.38(0.13})$ | $-3.66(0.53)$ | $-1.91(0.99)$ | $-2.63(0.50)$ |
| DS4 | $(41, 1137)$ | $-13506.05(0.32)$ | $0.03(0.53)$ | $\mathbf{0.46(0.43})$ | $-2.48(0.43)$ | $-0.47(1.21)$ | $-1.73(0.23)$ |
| DS5 | $(50, 378)$ | $-8271.26(0.39)$ | $-1.70(0.35)$ | $-5.69(0.48)$ | $-0.29(1.82)$ | $-4.01(0.28)$ | $\mathbf{0.94(2.08})$ |
| DS6 | $(50, 1133)$ | $-6745.31(0.55)$ | $-0.76(0.20)$ | $\mathbf{-0.32(0.35})$ | $-3.96(0.34)$ | $-3.26(0.60)$ | $-2.72(0.37)$ |
| DS7 | $(59, 1824)$ | $-37323.88(0.66)$ | $\mathbf{0.27(0.26})$ | $-0.24(0.17)$ | $-2.73(0.30)$ | $-2.82(0.31)$ | $-10.42(0.70)$ |
| DS8 | $(64, 1008)$ | $-8650.20(0.77)$ | $-0.82(0.27)$ | $\mathbf{0.47(0.64})$ | $-3.28(0.99)$ | $-4.95(0.47)$ | $-2.88(0.60)$ |
| DS9 | $(67, 955)$ | $-4072.66(0.53)$ | $-5.32(0.31)$ | $-4.12(0.46)$ | $\mathbf{-3.12(1.21})$ | $-5.79(0.74)$ | $-7.60(0.44)$ |
| DS10 | $(67, 1098)$ | $-10102.65(0.65)$ | $\mathbf{-0.88(0.20})$ | $-1.44(0.22)$ | $-5.38(0.42)$ | $-3.98(1.14)$ | $-6.82(0.49)$ |
| DS11 | $(71, 1082)$ | $-6272.57(0.68)$ | $-18.79(0.41)$ | $-16.28(0.46)$ | $\mathbf{-6.79(0.89})$ | $-7.31(0.71)$ | $-9.62(1.46)$ |
| COV | $(72, 3101)$ | $-7861.61(0.74)$ | $-39.08(0.58)$ | $\mathbf{-33.26(0.76})$ | $-611.84(1.80)$ | $-374.62(0.48)$ | $-214.25(0.42)$ |

*Table 7.* ***Estimated evidence lower bounds for variational inference methods (in nats).*** *ELBOs were estimated using importance sampling on 1,000 random samples from each variational distribution. Our VIPR methods beat the VBPI baseline on half of the datasets. Dataset names, method acronyms, and conditions match Table 6. Standard errors were estimated using 100 bootstrapped samples and are shown in parentheses.*

| DATA | $(N, M)$ | VBPI10 | VBPI20 | LOOR | REP | VIMCO |
|------|----------|--------|--------|------|-----|-------|
| DS1 | (27, 1949) | $-\mathbf{7157.99(0.15)}$ | $-7158.18(0.16)$ | $-7159.56(0.10)$ | $-7159.54(0.09)$ | $-7161.60(0.20)$ |
| DS2 | (29, 2520) | $-26573.03(0.28)$ | $-26573.60(0.30)$ | $-26569.56(0.06)$ | $-\mathbf{26569.50(0.08)}$ | $-26570.74(0.13)$ |
| DS3 | (36, 1812) | $-\mathbf{33793.96(0.20)}$ | $-33794.75(0.28)$ | $-33794.96(0.08)$ | $-33794.77(0.07)$ | $-33796.53(0.15)$ |
| DS4 | (41, 1137) | $-13541.39(13.12)$ | $-13613.68(22.18)$ | $-\mathbf{13512.54(0.11)}$ | $-13512.60(0.11)$ | $-13513.41(0.14)$ |
| DS5 | (50, 378) | $-8281.03(0.26)$ | $-8298.64(5.97)$ | $-\mathbf{8279.93(0.11)}$ | $-8280.35(0.11)$ | $-8282.03(0.17)$ |
| DS6 | (50,1133) | $-\mathbf{6751.77(0.22)}$ | $-6752.60(0.21)$ | $-6754.36(0.12)$ | $-6755.29(0.14)$ | $-6756.10(0.21)$ |
| DS7 | (59, 1824) | $-\mathbf{37331.12(0.22)}$ | $-37331.82(0.31)$ | $-37333.36(0.19)$ | $-37332.04(0.14)$ | $-37352.10(0.42)$ |
| DS8 | (64, 1008) | $-\mathbf{8657.78(0.30)}$ | $-8658.83(0.22)$ | $-8662.26(0.16)$ | $-8661.88(0.16)$ | $-8664.54(0.26)$ |
| DS9 | (67, 955) | $-4088.64(0.39)$ | $-4091.21(0.52)$ | $-\mathbf{4085.61(0.18)}$ | $-4087.25(0.20)$ | $-4090.52(0.22)$ |
| DS10 | (67, 1098) | $-\mathbf{10111.81(0.29)}$ | $-10112.80(0.28)$ | $-10114.76(0.15)$ | $-10115.16(0.16)$ | $-10119.70(0.26)$ |
| DS11 | (71, 1082) | $-6329.37(9.90)$ | $-6559.24(54.99)$ | $-6289.60(0.17)$ | $-6289.70(0.18)$ | $-6294.31(0.20)$ |
| COV | (72, 3101) | $-8100.96(109.62)$ | $-\mathbf{7913.84(0.93)}$ | $-8489.82(0.21)$ | $-8244.41(0.21)$ | $-8087.43(0.30)$ |

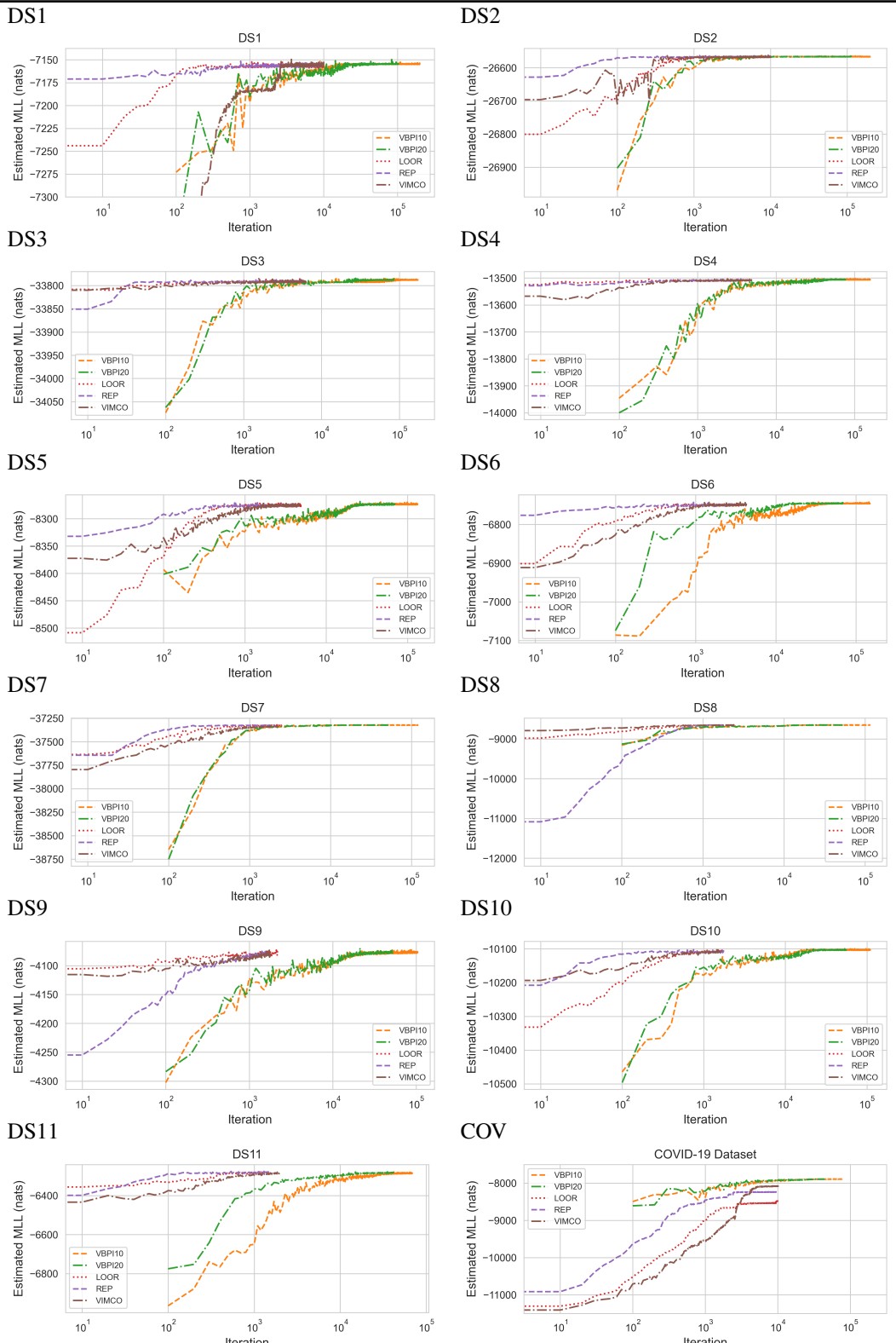

*Figure 5. **Trace plots for all datasets.*** Trace plot of estimated marginal log-likelihood vs. iteration number (*i.e.*, parameter update number). Marginal log-likelihood was estimated using 500 importance samples for VBPI and 50 importance samples for VIPR methods.

## B.5. Computational Complexity Results

We provide an additional plot for the results described on the MS datasets in Section 5.1. Computing the variational density of VBPI is linear in the number of taxa, but normalizing the SBN scales with the number of parameters. Therefore, the computational complexity of VBPI scales with the number of parameters. Figure 6 below further illustrates this by plotting the slope of a log-log curve against the number of taxa, where the y-axis represents the computational complexity of each algorithm.

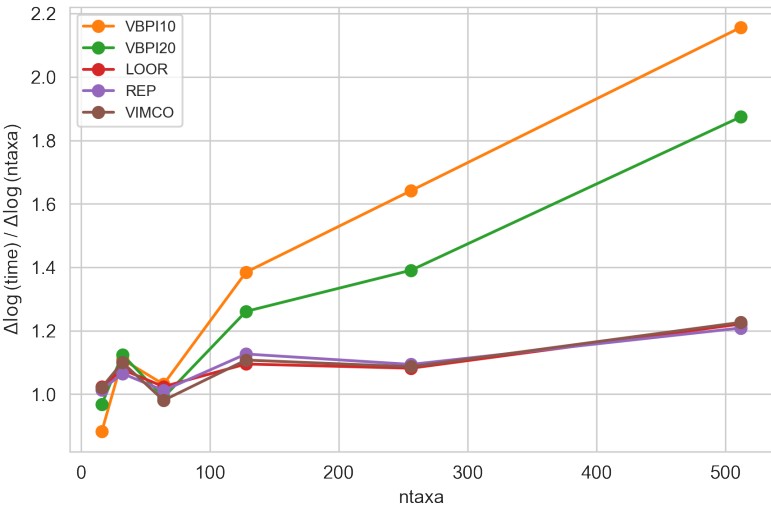

*Figure 6. **Slope of the logarithm of seconds-per-iteration vs. the logarithm of the number of taxa.** Each VI method was run for 1,000 iterations on subsets of the COVID-19 dataset. The y-axis corresponds to the computational complexity of the algorithm as a function of number of taxa (i.e., 1 corresponds to linear complexity, 2 corresponds to quadratic complexity, etc.)*

## C. Gradient Estimators for $q_\phi$

### C.1. The REINFORCE Estimator

Working from the definitions in Section 3.3.1, the leave-one-out REINFORCE estimator for VIPR is derived as follows:

$$\nabla_\phi L(\phi) \approx \frac{1}{K} \sum_{k=1}^{K} w^{(k)} \nabla_\phi \log q_\phi(\tau^{(k)}, \mathbf{t}^{(k)}), \tag{28}$$

$$w^{(k)} = f_\phi(\tau^{(k)}, \mathbf{t}^{(k)}) - \hat{f}^{(-k)}, \tag{29}$$

$$\hat{f}^{(-k)} = \frac{1}{K-1} \sum_{\ell \neq k} f_\phi(\tau^{(\ell)}, \mathbf{t}^{(\ell)}) \tag{30}$$

$$(\tau^{(k)}, \mathbf{t}^{(k)}) \sim q_\phi. \tag{31}$$

### C.2. The Reparameterization Trick

We continue the derivation of the reparameterization trick for VIPR. Working from Equation 9 in Section 3.3.2, we proceed by summing over the tree structures $\tau$ and then integrating over $\mathbb{Z}_\tau(\phi)$, the space of all values of $\mathbf{Z}$ that are consistent with $\tau$ given the parameters $\phi$. This yields the following:

$$L(\phi) = \sum_\tau \int_{\mathbf{Z} \in \mathbb{Z}_\tau(\phi)} \mathcal{N}(\mathbf{Z}; \mathbf{0}, I) \log \left( \frac{p(\mathbf{Y}, g_\phi(\mathbf{Z}))}{q_\phi(g_\phi(\mathbf{Z}))} \right) d\mathbf{Z}. \tag{32}$$

Note that the region of integration $\mathbb{Z}_\tau(\phi)$ depends upon $\phi$, so interchanging the integral and the gradient introduces some error due to the Leibniz integral rule. Nonetheless, we proceed with the interchange and approximate the full gradient as follows:

$$\nabla_\phi L(\phi) \approx \mathbb{E}_{\mathbf{Z}} \left[ \nabla_\phi \log \left( \frac{p(\mathbf{Y}, g_\phi(\mathbf{Z}))}{q_\phi(g_\phi(\mathbf{Z}))} \right) \right]. \tag{33}$$

Finally, we define a *biased* estimate of $\nabla_\phi L(\phi)$ as the following:

$$\widehat{\nabla}_\phi L(\phi) \approx \frac{1}{K} \sum_{k=1}^{K} \nabla_\phi \log \left( \frac{p(\mathbf{Y}, g_\phi(\mathbf{Z}^{(k)}))}{q_\phi(g_\phi(\mathbf{Z}^{(k)}))} \right) \tag{34}$$

$$\mathbf{Z}^{(k)} \sim \mathcal{N}(\cdot; \mathbf{0}, I). \tag{35}$$

As with the LOOR estimator, the gradient $\nabla_\phi \log \left( \frac{p(\mathbf{Y}, g_\phi(\mathbf{Z}^{(k)}))}{q_\phi(g_\phi(\mathbf{Z}^{(k)}))} \right)$ can be calculated using automatic differentiation software such as Autograd (Maclaurin et al., 2015) or PyTorch (Paszke et al., 2019).

### C.3. The VIMCO Estimator

We derive the VIMCO Estimator for VIPR described in Section 3.3.3. For our model, the $k$-sample ELBO (Mnih and Rezende, 2016) is defined as follows:

$$L_K(\phi) = \mathbb{E}_{q_\phi} \left[ \log \left( \frac{1}{K} \sum_{i=1}^{K} \frac{p(\tau^{(k)}, \mathbf{t}^{(k)}, \mathbf{Y}^{(\text{ob})})}{q_\phi(\tau^{(k)}, \mathbf{t}^{(k)})} \right) \right]. \tag{36}$$

Here, $(\tau^{(k)}, \mathbf{t}^{(k)}) \sim q_\phi$ for $k = 1, \ldots, K$. This is the objective function used by Zhang and Matsen IV (2024) to perform VBPI. When using the $K$-sample ELBO objective from Equation (36), the VIMCO estimator is an analogous gradient estimator to the LOOR estimator for the single-sample ELBO, and is defined as follows:

$$\nabla_\phi L_K(\phi) \approx \sum_{k=1}^{K} \left( \hat{L}_K^{(-k)}(\phi) - \tilde{w}^{(k)} \right) \nabla_\phi \log q_\phi(\tau^{(k)}, \mathbf{t}^{(k)}) \tag{37}$$

$$\tilde{w}^{(k)} = \frac{f_\phi(\tau^{(k)}, \mathbf{t}^{(k)})}{\sum_{\ell=1}^{K} f_\phi(\tau^{(\ell)}, \mathbf{t}^{(\ell)})} \tag{38}$$

$$\hat{L}_K^{(-k)}(\phi) = \hat{L}_K(\phi) - \log \frac{1}{K} \left( \sum_{\ell \neq k} f_\phi(\tau^{(\ell)}, \mathbf{t}^{(\ell)}) + \hat{f}_\phi^{(-\ell)} \right) \tag{39}$$

$$\hat{L}_K(\phi) = \log \left( \frac{1}{K} \sum_{k=1}^{K} f_\phi(\tau^{(k)}, \mathbf{t}^{(k)}) \right) \tag{40}$$

$$\hat{f}^{(-\ell)} = \frac{1}{K-1} \sum_{j \neq \ell} f_\phi(\tau^{(j)}, \mathbf{t}^{(j)}) \tag{41}$$

$$(\tau^{(k)}, \mathbf{t}^{(k)}) \sim q_\phi. \tag{42}$$

