# OpenReview forum: "Variational Phylogenetic Inference with Products over Bipartitions"
_ICML.cc/2025/Conference — ICML 2025 poster_

### Official Review · Reviewer_uJ3o · 2025-03-10

**Overall Recommendation:** 3

**Summary:**

This paper targets variational inference of ultrametric phylogenetic trees and proposed a method called VIPR.
Although many efforts have been paid in the field of machine learning based variational phylogenetic infernece, very few researchers consider this on ultrametric trees.
VIPR sample a ultrametric phylogenetic tree by executing single linkage clustering on a distance matrix which is learnable and parametrized by log-normal distribution.
A main contribution of this paper is the density formula of the abovementioned tree distribution (Proposition 1), and this allows training VIPR with gradient-based methods.
The authors validate the effectiveness of VIPR on DS1-11 and Cov-2 benchmarks and compare VIPR to the sota method VBPI.

## update after rebuttal

First of all, I would like to thank the authors for their detailed response that addresses my questions well. I have one further question on the convergence speed of VIPR. The author shows that it converges faster than other baselines (e.g., VBPI). However, this is a design choice in VBPI that uses the anneal schedule to encourage exploration over the tree space. Or in other words, there is a trade-off between the convergence speed to a good likelihood value of trees and the coverage of the high posterior region. I would expect the convergence speed of VIPR to slow down a bit with a similar annealing schedule, but end up with a better coverage over posterior trees (this would potentially improve the performance of VIPR in terms of marginal likelihood estimation). It would also be interesting to show how close is the approximate tree topology distribution provided by VIPR to the ground truth posterior from MCMC.

Overall, I really like the idea of constructing a variational distribution over ultrametric trees from a distribution of pairwise distances through single-linkage clustering. It would be interesting to explore how to make it flexible enough for complicated tree posterior, which seems a bit challenging, as it is nontrivial how the correlation between pairwise distance would translate into the correlation between tree topologies. That said, I have updated my review accordingly.

**Claims And Evidence:**

Although this paper presents a sound methodology for VIPR, its potential advantages (e.g., inference accuracy or speed) are not revealed by the experiments.
For the inference accuracy, the likelihoods of VIPR's inferred trees lagged behind those from VBPI, as suggested by the Figure 3 (b,c) and Table 2.
For the inference speed, no results directly reports the computation time of different methods.

**Essential References Not Discussed:**

- The claim "The VBPI baseline requires MCMC runs to determine likely subsplits (i.e., evolutionary branching events)." in Sec 4.2 is not accurate. A notable progress of VBPI is ARTree (NeurIPS 2023; https://arxiv.org/abs/2310.09553), which does not rely on subsplits and should be discussed.
- Phyloformer (https://www.biorxiv.org/content/10.1101/2024.06.17.599404v1) constructs a phylogenetic tree with a neighbor joining algorithm on pairwise representations. This idea is similar to VIPR.

**Experimental Designs Or Analyses:**

- As the variational family in VBPI is applied to the general class of additive phylogenetic trees, the authors should clearly explain how they use VBPI to infer ultrametric trees.
- GeoPhy (NeurIPS 2023; https://arxiv.org/pdf/2307.03675) considered a NJ algorithm on distance matrix for constructing phylogenetic trees, similarly to VIPR. It should be considered as an important baseline in terms of inference speed and accuracy.
- Figure 3(c) (VBPI is better) contradicts with Table 2 (VIPR-VIMCO is better).

**Methods And Evaluation Criteria:**

The proposed methods and/or evaluation criteria make sense.

**Other Comments Or Suggestions:**

There is one typo, Line 32: PhylogGFN.

**Other Strengths And Weaknesses:**

Weaknesses:
- The title page violates the ICML format (one-column titles and missing author information). I suggest the authors cut down the length of Results and Discussion to creating some space for the title page.
- I think ultrametirc trees should be defined by "the leaves of the trees are all equidistant from the root", and the authors' definition in Sec 2.1 seems somewhat misleading.
- The authors does not clearly explain the $N_e$ in the prior distribution in Sec 2.3.

**Questions For Authors:**

I have no other questions fo the authors.

**Relation To Broader Scientific Literature:**

Many prior works considered variational phylogenetic inference, e.g., VBPI, GeoPhy, but few of them considered inferring ultrametric ones.
This paper makes contribution in this sense.

**Theoretical Claims:**

I've read the proof for Proposition 1 but not checked it carefully.

---

> ### Author Rebuttal · Authors · 2025-04-01
>
> Thank you for your insightful review: see our responses below.
>
> *For inference accuracy, VIPR's trees' likelihoods lagged behind VBPI. For inference speed, no results report computation time.*
>
> The aspect where VIPR shines is in time (or number of parameter updates) to attain an approximation error, as shown in Figure 5 of Appendix B. For example, on DS1 VIPR-LOOR achieves an estimated marginal log-likelihood (MLL) of -7175 after \~100 iterations, while VBPI10 takes \~1000. For DS5, VIPR-LOOR achieves an MLL of -8300 after \~200 iterations, while VBPI10 takes \~5000.
>
> Figure 5 compares iterations, and we agree that we should also report computation time. Thus, we ran all VI methods on simulated datasets with varying numbers of taxa for 1,000 iterations and reported computation time (see our response to Reviewer 1 for the simulation procedure).
>
> Seconds/1,000 iterations:
>
> taxa | VBPI10 | VBPI20 | LOOR | REP | VIMCO
> ---|---|---|---|---|---
> 8 | 24 | 45 | 55 | 74 | 55
> 16 | 43 | 88 | 110 | 150 | 112
> 32 | 94 | 192 | 234 | 314 | 240
> 64 | 192 | 381 | 475 | 633 | 473
> 128 | 500 | 913 | 1,016 | 1,383 | 1,018
> 256 | 1,560 | 2,395 | 2,150 | 2,952 | 2,162
> 512 | 6,958 | 8,780 | 5,014 | 6,822 | 5,060
>
> One iteration is one parameter update. Our method is \~ twice as slow as VBPI per iteration for 8 taxa, but it scales better and outperforms VBPI for 512 taxa. We will add the results to Figure 4 and the Appendix.
>
> We improved our code since submission (see our response to Reviewer 1). VIPR's primary computational bottleneck is now the phylogenetic likelihood, which takes $\mathcal{O}(NM)$ time.
>
> *How is VBPI used for ultrametric trees?*
>
> Zhang and Matsen IV [ICLR 2019] applies to general additive phylogenetic trees, but the follow-up paper [Zhang and Matsen IV, JMLR2024] extends the approach to ultrametric trees in Sections 6 and 7. The github repository (https://github.com/zcrabbit/vbpi-torch) contains code for ultrametric trees (in the directory "rooted") that we use in our experiments. We will make this more clear in the manuscript by providing a Github repository link and referencing the sections within the JMLR paper.
>
> *GeoPhy should be considered*
>
> GeoPhy is similar to VIPR in that it uses a tree construction algorithm on a distance matrix, but it is used for unrooted trees while we focus on ultrametric trees.
>
> *Figure 3(c) (VBPI is better) contradicts with Table 2 (VIPR-VIMCO is better).*
>
> Figure 3(c) reports the marginal log-likelihood, but Table 2 reports the ELBO. For the COVID dataset, VBPI is better in marginal log-likelihood, and VIPR-VIMCO is better in ELBO.
>
> *ARTree should be discussed.*
>
> We will change Section 4.2 (line 264, column 2): "We compare VIPR to the VBPI algorithm as implemented by Zhang and Matsen IV (2024), which uses MCMC runs to determine likely subsplits in an SBN."
>
> We will also add ARTree to the introduction (line 26, column 2): "For example, ViaPhy (Koptagel et al., 2022) uses a gradient-free variational inference approach and directly sample from the Jukes and Cantor (1969) model, GeoPhy (Mimori and Hamada, 2023) uses a distance-based metric in hyperbolic space to construct unrooted phylogenetic trees, and ARTree (Xie and Zhang, 2023) uses graph neural networks to construct a deep autoregressive model for variational inference over phylogenetic tree structures."
>
> *Phyloformer is similar to VIPR*
>
> We will mention Phyloformer in the introduction: see our response to Reviewer 1.
>
> *The title page violates the ICML format*
>
> Thank you for catching the title formatting error. This arose due to a copy/paste mistake. As Reviewer 2 noted, our Figures are not too space-hungry, and we could resolve this by cutting down the length of the Results and Discussion as you suggest, and improving the location of the Figures.
>
> *Ultrametirc trees should be (re)defined*
>
> We will adopt the suggested definition (line 74, column 1):
>
> "We focus on ultrametric trees, in which the leaves of the trees are all equidistant from the root. We denote our ultrametric trees with a rooted, binary tree topology tau and a set of coalescent times ..."
>
> *The $N_e$ in the prior distribution is unclear*
>
> We expanded section 2.3:
>
> "We use the Kingman coalescent (Kingman 1982) as the prior distribution on the trees. This coalescent process proceeds backward in time with exponentially distributed inter-event intervals, and coalescent events occurring at rate $\lambda_k = \binom{k}{2}/N_e$, where $k$ is the number of taxa and $N_e$ is the effective population size, a parameter which governs the rate at which species coalesce. We fix $N_e = 5$ in our experiments. At each coalescent event, a pair of taxa are chosen to coalesce into a single taxon uniformly at random over all pairs of taxa. ... "
>
> We fixed the "PhylogGFN" typo, thank you for catching this.

---

### Official Review · Reviewer_R16S · 2025-03-12

**Overall Recommendation:** 3

**Summary:**

This paper proposes a variational Bayesian phylogenetic tree analysis method using a matrix representation of tree structures. Phylogenetic tree analysis is one of the important analytical techniques used to estimate the developmental process and diffusion pathways of a target, and is more and more in demand in formulating future preventive measures, for example, for recent infectious disease pandemics. Conventional Bayesian phylogenetic tree analysis faces several challenges. One is the use of Markov chain Monte Carlo methods in many models and algorithms, whose efficiency, both theoretically and empirically, is not yet clear. Another is that many of them do not explicitly include the fusion time in the phylogenetic tree in their models, and as a result, those methods cannot properly capture the ultrametric nature of the tree structure. As a way to solve these two problems, this paper proposes a model and its inference method that can properly reflect ultrametric by explicitly modeling coalescence time using a tree structure representation that has high affinity to variational methods. The effectiveness of the proposed method is demonstrated with 7 data frequently used in many recent Bayesian phylogenetic analyses and the more practical SARS-Cov-19 data.

**Claims And Evidence:**

The main claim of this paper is that using a matrix representation for the representation of tree structures has two benefits: (1) ultrametric measures can be captured and (2) a differentiable variational representation that avoids MCMC sampling, whose theoretical analysis and empirical goodness for mixing time is not yet clearly known.

One minor concern is that a similar tree-structured matrix representation has been studied independently in another paper [Bouckaert2024] very recently. However, the authors, in fairness, identify differences and improvements over prior work in Section 2.5.

**Essential References Not Discussed:**

This paper provides a comprehensive discussion of Bayesian phylogenetic tree analysis, from its historical development to the latest developments in recent years. In particular, the relationship between the proposed methods and the challenges and room for improvement are carefully discussed in fairness to recent related research.

**Experimental Designs Or Analyses:**

Experiments are conducted on seven datasets that have been used expressively as benchmark data in recent Bayesian phylogenetic tree analyses, as well as on the SARS-Cov-19 data for more practical applications. The evaluation by the marginalized likelihood and ELBO also reflects recent trends, and the experts feel that the improvement in performance is clearly reported.

The comparison method seems convincing enough, as it is extremely up-to-date. On the other hand, as discussed in the “Methods” section, a quantitative comparison with another related study A might have strengthened its persuasiveness.

**Methods And Evaluation Criteria:**

This paper uses a variational phylogenetic tree representation that utilizes a matrix representation of the tree structure and derives an inference algorithm using three choices of loss functions. Through experiments, the proposed method is compared to previous state-of-the-art phylogenetic tree analysis methods (including the most related and recent one [Zhang&Matsen, JMLR2024]). The evaluation criteria used are the marginalized likelihood and ELBO in terms of learning and prediction performance.

One minor concern to me is whether the ablation study made it difficult to quantitatively assess the improvement of the proposed method from the other related study, [Bouckaert2024]. Would it be difficult to see the reduction in performance when restricted to only restricted trees, as in Literature A, within the framework of the proposed method?

**Other Comments Or Suggestions:**

I was just a little concerned as to whether the title conforms to the format specified for the conference. Considering the margins involved in some of the figures, this is not overly space-hungry and may not be a problem for the draft stage for peer review. However, it may be a good idea to have it corrected in the camera-ready version if accepted.

**Other Strengths And Weaknesses:**

(Editing)

**Questions For Authors:**

One minor concern to me is whether the ablation study made it difficult to quantitatively assess the improvement of the proposed method from the other related study, [Bouckaert2024]. Would it be difficult to see the reduction in performance when restricted to only restricted trees, as in Literature A, within the framework of the proposed method?

The key theoretical contribution of this paper is that through the matrix representation of the tree structure, the variational distribution obtains an easy-to-handle closed-form representation.
I may not yet properly understand the empirical benefit of this theoretical result; as shown in Appendix 1, I can see that this representation does indeed lead to an easy-to-handle closed-form expression. On the other hand, it is not easy to intuitively understand what improvement this has over the variational distribution of the standard mean-field approximation. Any help from the authors in this regard would be greatly appreciated.

**Relation To Broader Scientific Literature:**

As evoked by the recent pandemics, phylogenetic tree analysis is one of the machine learning tasks that has received particular attention in recent years. This paper is not intended to bring any new insights from a scientific point of view, but this technology is expected to contribute to the development of computational biology through the development of general-purpose machine learning.

**Theoretical Claims:**

The key theoretical contribution of this paper is that through the matrix representation of the tree structure, the variational distribution obtains an easy-to-handle closed-form representation, as described in Proposition 1.

The key theoretical contribution of this paper is that through the matrix representation of the tree structure, the variational distribution obtains an easy-to-handle closed-form representation.
I may not yet properly understand the empirical benefit of this theoretical result; as shown in Appendix 1, I can see that this representation does indeed lead to an easy-to-handle closed-form expression. On the other hand, it is not easy to intuitively understand what improvement this has over the variational distribution of the standard mean-field approximation. Any help from the authors in this regard would be greatly appreciated.

---

> ### Author Rebuttal · Authors · 2025-04-01
>
> We appreciate the thoughtful suggestions below, and hope that we have addressed your comments sufficiently.
>
> *One minor concern is that a similar tree-structured matrix representation has been studied independently in another paper [Bouckaert2024] very recently.*
>
> As we discuss in our literature review, we have two primary significant improvements compared to [Bouckaert2024]: 1) we use optimization-based VI to maximize the ELBO, and 2) we provide a closed-form density in Proposition 1. Using this density means we do not have to restrict the tree space. In addition, our framework allows us to use a relatively general class of variational distributions for the pairwise distances, whereas Bouckaert require log-normal distributions in order to incorporate a covariance matrix $\Sigma$.
>
> *Would it be difficult to see the reduction in performance when restricted to only restricted trees within the framework of the proposed method?*
>
> We believe that this comment refers to selecting an ordering of taxa similarly to [Bouckaert2024], and then re-running VIPR while restricted to the "cube space" from [Bouckaert2024] consistent with the ordering. This is an excellent idea to investigate the effect of restricting tree space.
>
> We have now constructed the maximum clade credibility (MCC) tree from BEAST using our gold standard MCMC run, selected an order from the MCC tree, and then calculated the percentage of tree topologies from the BEAST gold standard that are within the "cube space" implied by this ordering. This process estimates the percentage of the posterior that is impossible to reach using the restricted tree space from [Bouckaert2024]:
>
> DS | \% of MCMC trees outside cube space
> ---|---
> 1 | 29.2
> 2 | 15.2
> 3 | 76.8
> 4 | 79.7
> 5 | 98.0
> 6 | 94.7
> 7 | 69.9
> 8 | 42.7
> 9 | 99.9
> 10 | 84.6
> 11 | 99.9
> COV| 99.9
>
> These results are striking, but [Bouckaert2024] mentions that CubeVB may struggle on high-entropy posteriors in their discussion. We will add this Table and discussion to the Appendix of the camera-ready.
>
> *It is not easy to intuitively understand what improvement VIPR has over the variational distribution of the standard mean-field approximation.*
>
> One of the key challenges for variational inference over phylogenetic trees is that using a mean-field approximation is not straightforward. There are two main reasons for this.
>
> First, we can only apply a mean-field approximation after decomposing the distribution of the tree as a product over cliques of random variables. There is no standard way of doing this for trees. In Matsen IV (2024) this is done by forming a subsplit Bayesian network with one node per subtree appearing in the MCMC samples used to initialize the support. Our novel decomposition (Proposition 1) is another way of decomposing the distribution of the tree as a product over coalescent times. This results in $\mathcal{O}(N^2)$ parameters, improving upon the worst-case performance of Matsen IV (2024), in which the number of parameters could be super-exponential in the worst case. We then use our novel decomposition for the mean-field approximation (using state-of-the-art techniques for gradient evaluation and optimization: autograd, VIMCO, REINFORCE and the Reparameterization Trick).
>
> Second, for ultrametric trees we cannot assume independence between coalescent times, lest the resulting tree violate the ultrametric constraint. To overcome this challenge, we form our our variational family to approximate the matrix of pairwise coalescent times (the matrix bold T). We then map T to ultrametric trees using single-linkage clustering.
>
> *A quantitative comparison with another related study might have strengthened its persuasiveness.*
>
> We chose VBPI for a baseline comparison because it was the only VI-based method for ultrametric trees that we are aware of in the literature. We did not include [Bouckaert2024] because it does not rely on optimization, so we could not include it in our trace plots of marginal log-likelihood vs iteration number. As you have suggested, restricting our method (and the BEAST gold-standard) to the same restricted tree space as [Bouckaert2024] is a valuable experiment to isolate the effects of optimization versus unrestricted tree space, and we aim to complete this experiment in a follow-up paper.
>
> Thank you for catching the title formatting error&mdash;we have now corrected it.

---

### Official Review · Reviewer_en52 · 2025-03-12

**Overall Recommendation:** 4

**Summary:**

This paper introduced a new method, VIPR, for phylogenetic inference. This new method greatly improves the computational efficiency without sacrificing accuracy compared with the traditional MCMC based method. The new method derives a closed-form density of the distribution over the entire tree space based on coalescent times and single-linkage clustering. This study proposed a new variational distribution based on coalescent time and single linkage clustering, which makes the computation more efficient. Experiments on benchmark dataset and one empirical dataset shows comparable accuracy and improved computational efficiency.

**Claims And Evidence:**

This paper claims that the new method VIPR relaxes the dependency on MCMC subroutines and achieves better efficiency on phylogenetic inference. Despite lower computational complexity, this new method achieves comparable accuracy with the golden standard Bayesian phylogenetic inference methods.
The claims are supported by experiments on benchmark datasets and one empirical dataset, SARS-CoV-2, comparison across baselines, including BEAST, the gold standard MCMC-based method for phylogenetic inference, and approximate the true posterior distribution, and VBPI, a recent variational inference method for phylogenetic inference using subplot Bayesian networks but still rely on MCMC for tree sampling.
The experiment was conducted on 11 standard benchmark datasets with a wide range of taxa numbers and sequence length, and one empirical dataset, SARS-CoV-2, to evaluate real-world dataset with rapid evolving speed.
The results show that VIPR performs comparable with two baselines on MLLs and ELBO. The running time of the experiments shows that VIPR has a time complexity of roughly O(N^2).
One minor concern is that VBPI shows roughly the same computational complexity on empirical dataset. Could the author elaborate more on how does the parameter numbers influence the time complexity of VBPI? Why VIPR should have a lower computational complexity?

**Essential References Not Discussed:**

To achieve a comprehensive landscape of phylogenetic inference studies, the author should consider to discuss other tree inference methods, such as maximum likelihood methods and distance based heuristics.

**Experimental Designs Or Analyses:**

The experiment design is reasonable.
The datasets covers a relatively wide range of difficulty levels. Including 11 standard benchmark datasets and 1 empirical dataset.
The method performance is compared against the MCMC-based golden standard method, BEAST, and a latest variational bayesian phylogenetic inference method, VBPI.
The metrics used for evaluation are valid.

Could the author add more baseline methods such as faster heuristic methods like RAxML?
The experiment does not cover the uncertainty estimation.

**Methods And Evaluation Criteria:**

VIPR is proposed based on the previous studies of Bouckaert (2024), and Zhang and Matsen IV (2024) with improvements on the scalability and computational efficiency.
VIPR does not rely on MCMC sampling like the other traditional methods. It directly models the distribution over the tree space. Compared with VBPI, this new method uses a variational distribution over distance matrix. This derives a differentiable variational distribution over the tree space, makes it possible to apply efficient gradient estimation for faster and more stable inference.
VBPI directly optimizes the coalescent times/branch lengths, relaxes the limitation of to Bouckaert (2024) method matrix representation approach on the ability of tree representation.
The methods are evaluated on benchmark datasets and empirical dataset. The benchmark datasets covers a wide range of taxa numbers and sequence length, representing a range of complexity.

VIPR assume a log-normal variational distribution. What could be the impact of this assumption? Any chance to relax this assumption to achieve better flexibility on the inference?
Another minor issue is that the sequence divergence is not included for the datasets. It would be helpful to get a rough sense of how difficult are those datasets and what is the impact on the method performance.
The author can also consider to include simulated dataset with better control on tree depth, sequence divergence, mutation rates, etc.
The paper only considered Jukes-Cantor model, which may be over-simplified. Could the author consider more complex evolutionary models such as GTR?

**Other Comments Or Suggestions:**

I have listed my comments in previous sections.

**Other Strengths And Weaknesses:**

This new method shows good novelty. VIPR addresses an important limitation of previous methods, removes dependency of MCMC sampling. VIPR makes the Bayesian tree inference more scalable for larger datasets.
The paper is well-structured, with a clear motivation, theoretical proof, and experimental results.
Overall, the paper shows a strong contribution to Bayesian phylogenetic inference method.

The paper is relatively simplified on the theoretical assumptions. Please consider to expand the methods to more complex evolutional model to better fit the real-world applications. The paper can be stronger by adding discussion on uncertainty estimation, comparison with other mainstream phylogenetic inference methods, such as RAxML, neighbor-joining, and Vaiphy.

**Questions For Authors:**

I'm curious about the following questions:
1. How would VIPR handle non-ultrametric trees?
2. How robust is VIPR when handling noisy or high divergent dataset?
3. What is the limitation of VIPR on extreme large dataset, for examples dataset with 100+ taxa?

**Relation To Broader Scientific Literature:**

VIPR improves the previous methods introduced by Bouckaert (2024), and Zhang and Matsen IV (2024) with better scalability and computational efficiency. Current MCMC-based methods are limited to small datasets <100 taxas due to high computational cost. With the new method, it could enable Bayesian method on much larger dataset.
This new method also enables better integration with machine learning pipelines. Traditional phylogenetic inference methods are not differentiable. VIPR provides a differentiable method that is compatible with deep learning pipelines.

**Theoretical Claims:**

This paper defines the tree space and shows its variational distribution covers the entire tree space. VIPR variational family enables gradient-based optimization. Proposition 1 shows a closed-form solution for the density function of trees. The probability density function over trees looks good. The derivations of gradient estimators look correct.
The theoretical claims are mostly valid and proofs look correct.

---

> ### Author Rebuttal · Authors · 2025-04-01
>
> Thank you for your thoughtful review: please find detailed responses below.
>
> *How do parameter numbers influence the time complexity of VBPI?*
>
> In [Zhang and Matsen, JMLR2024], as the number of taxa grows, the number of parameters grows with the number of trees in the SBN. There is no closed form for the number of trees; it depends on the MCMC algorithm and posterior concentration. We empirically calculated the number of parameters in VBPI vs the number of taxa on datasets simulated with "ms" [Hudson 2002] with 1,000 sites.
>
> \# of tree structure parameters:
> taxa | VBPI | VIPR |
> ---|---|---
> 8 | 4 | 56
> 16 | 44 | 240
> 32 | 55 | 992
> 64 | 3,826 | 4,032
> 128 | 29,939 | 16,256
> 256 | 127,217 | 65,280
> 512 | 319,533 | 261,632
>
> Computing the variational density of VBPI is linear in the number of taxa, but normalizing the SBN scales with the number of parameters. We will add this experiment to the Appendix of the camera-ready copy.
>
> *Why should VIPR have a lower computational complexity?*
>
> In our experiments, VIPR attains accurate marginal log-likelihoods estimates in fewer parameter updates than VBPI (Appendix B, Figure 5). VIPR has O($N^2$) parameters, and the number of parameters in VBPI can be larger than that if the SBN support is large (see Table above).
>
> We improved our code using *scipy.cluster.hierarchy.linkage* and streamlined our phylogenetic likelihood function. We performed new speed comparisons for all methods on simulated datasets with varying numbers of taxa. See our response to Reviewer 3 for results.
>
> *What is the impact of log-normal branch-lengths? Any way to relax this?*
>
> After running BEAST, we plotted histograms of pairwise log-coalescent times across sampled trees for some of the datasets. In most cases these histograms looked normal, motivating our log-normal branch lengths. We will include a some of these histograms as a supplement. VIPR can incorporate any branch length distribution with continuously differentiable density. We will consider flexible branch-length distributions in future work.
>
> *Sequence divergence is not included.*
>
> We calculated pairwise Hamming distances between each taxa for each dataset (dropping sites with missingness). Values in parentheses are standard deviations:
>
> DS | Hamming distance/\#sites
> ---|---
> 1 | .040(.017)
> 2 | .214(.057)
> 3 | .230(.051)
> 4 | .138(.055)
> 5 | .192(.041)
> 6 | .056(.029)
> 7 | .203(.069)
> 8 | .082(.031)
> 9 | .025(.014)
> 10 | .070(.026)
> 11 | .082(0.053)
> COV | .008(0.003)
>
> We will add this Table to Appendix B in the camera-ready.
>
> *Include simulated datasets with better control on tree depth, sequence divergence, mutation rates, etc.*
>
> This simulation is an excellent idea. We aim to do this in a follow-up paper.
>
> *Jukes-Cantor may be over-simplified vs. complex evolutionary models*
>
> We agree that Jukes-Cantor is a simplified assumption. We aim to include K2P [Kimura 1980] and GTR [Rodriguez 1990] in a follow-up paper. (Note that Zhang and Matsen JMLR2024 only consider JC.)
>
> *Add more baseline methods like RAxML, neighbour-joining, and Vaiphy? The experiment does not cover uncertainty estimation.*
>
> These methods do not apply specifically to variational inference over ultrametric trees. RAxML and Neighbour-joining do not provide estimates of the marginal likelihood. Vaiphy is for multifurcating trees. We have already mentioned VaiPhy and we will add more about non-Bayesian methods in the introduction:
>
> "Phylogenetic inference can also be performed using non-Bayesian methods, including RAxML, Neighbour-joining, and Phyloformer. Phyloformer uses deep learning to construct pairwise representations of evolutionary distances between taxa. Phyloformer then uses pairwise distances to construct a  tree using a neighbor-joining algorithm similar to the method described here. Non-Bayesian methods do not provide estimates of marginal likelihood, which are useful for model selection."
>
> Regarding uncertainty estimation, we aim to add posterior predictive checks of tree length and clade support on simulated data to the Appendix for the camera-ready.
>
> *How would VIPR handle non-ultrametric trees?*
>
> A natural approach for non-ultrametric trees with our framework is to extend our strict clock models to relaxed clock models. Preliminary calculations suggest it is possible to do so at the cost of roughly twice as many variational parameters compared to the VIPR variational family.
>
> *How robust is VIPR when handling noisy or high divergent dataset?*
>
> VIPR struggled most with the COVID-19 dataset, where genomes varied relatively little (see Hamming distance Table above). We assume that highly divergent datasets would also be challenging. Future studies can quantify how VI methods such as VIPR are affected by noisy or divergent datasets.
>
> *What is the limitation of VIPR on dataset with 100+ taxa?*
>
> VIPR's empirical computation time per iteration is approximately linear in the number of taxa (see response to Reviewer 3). Future work can apply VIPR to larger datasets.

---

### Decision · Program_Chairs · 2025-05-01

**Decision:**

Accept (poster)

**Comment:**

This article propose a variational inference approach to the challenging problem of Bayesian phylogenetic inference. The key idea is proposing a variational family related to trees from hierarchical clustering, and because the density admits a simple closed form, it avoids much of the costly computation that existing methods are subject to. Reviewers all lean positive and while they did not discuss after some prompting, my own read agrees with their positive sentiment and find that this is a quality paper and the ideas will be interesting to the ICML readership.